# HINT: Hierarchical Interaction Modeling for Autoregressive Multi-Human Motion Generation

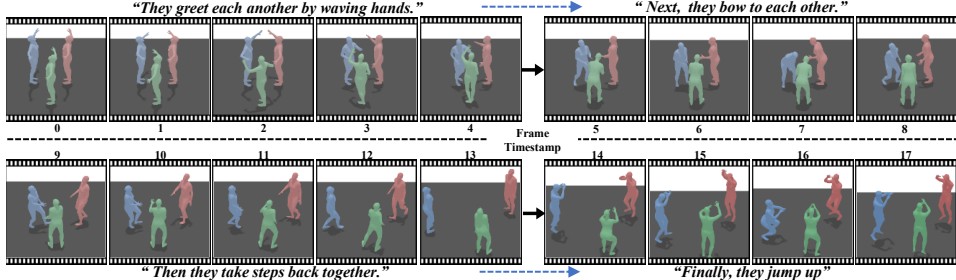

Figure 1: **Visualization of three-human motion generation results of HINT.** By continuously updating the text guidance, HINT can autoregressively generate coherent, plausible human motions.

## Abstract

Text-driven multi-human motion generation with complex interactions remains a challenging problem. Despite progress in performance, existing offline methods that generate fixed-length motions with a fixed number of agents, are inherently limited in handling long or variable text, and varying agent counts. These limitations naturally encourage autoregressive formulations, which predict future motions step by step conditioned on all past trajectories and current text guidance. In this work, we introduce **HINT**, the first autoregressive framework for multi-human motion generation with **H**ierarchical **INT**eraction modeling in diffusion. First, HINT leverages a disentangled motion representation within a canonicalized latent space, decoupling local motion semantics from inter-person interactions. This design facilitates direct adaptation to varying numbers of human participants without requiring additional refinement. Second, HINT adopts a sliding-window strategy for efficient online generation, and aggregates local within-window and global cross-window conditions to capture past human history, inter-person dependencies, and align with text guidance. This strategy not only enables fine-grained interaction modeling within each window but also preserves long-horizon coherence across all the long sequence. Extensive experiments on public benchmarks demonstrate that HINT matches the performance of strong offline models and surpasses autoregressive baselines. Notably, on InterHuman, HINT achieves an FID of 3.100, significantly improving over the previous state-of-the-art score of 5.154.

## 1 Introduction

Human motion generation shows diverse applications spanning character animation (Petrovich et al., 2022), human-robot interaction (Sahili et al., 2025), virtual reality (Chen et al., 2024), and content creation (Tevet et al., 2022; Guo et al., 2022). Recently, text-driven approaches (Javed et al., 2024; Liang et al., 2024; Zhao et al., 2024) have received growing attention, as they allow natural language to serve as a human-friendly control for generating semantically aligned human trajectories. Beyond the single-human setting (Tevet et al., 2022; Zhang et al., 2024b; Barquero et al., 2024), generating realistic, diverse, and controllable interactions for multiple humans remains highly challenging.

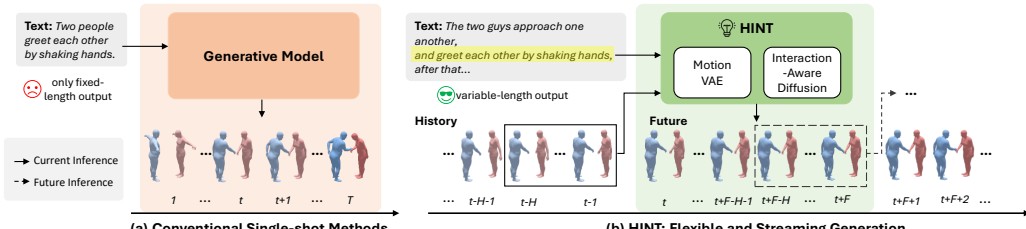

Figure 2: **Architecture Comparison.** (a) **Conventional Single-shot Methods**: Existing approaches (e.g., InterGen, in2IN, InterMask) generate motion sequences in a single shot with fixed length. (b) **HINT:** Our framework integrates autoregressive and diffusion modeling to support streaming generation. Within a sliding window, the Interaction-Aware Diffusion leverages history and text to progressively synthesize future motions, thereby supporting open-ended, variable-length generation.

Existing approaches (Javed et al., 2024; Liang et al., 2024) are offline frameworks that generate motions of a fixed frame length and a fixed number of agents, as shown in Fig. 2 (a). While effective for short sequences, these methods are inherently limited in handling variable-length natural language descriptions, dynamic interaction patterns, and varying agent counts. Moreover, they often fail to capture long-range dependencies across extended motion sequences, leading to incoherent or repetitive behaviors. These challenges naturally call for autoregressive formulations, as shown in Fig. 2 (b), where future motions are generated step by step conditioned on both past trajectories and textual instructions. Yet, autoregressive models in this domain remain underexplored, particularly for the multi-human setting with complex interactions.

In this work, we propose **HINT**, the first autoregressive diffusion-based framework for multi-human motion generation with hierarchical interaction modeling in diffusion. HINT presents two novel contributions. **First**, Canonicalized Latent Space, which encodes each human's motion in its own local coordinate system, rather than encoding all agents in world coordinates (Liang et al., 2024). Prior approaches (Liang et al., 2024; Javed et al., 2024; Ruiz-Ponce et al., 2024) typically adopt such joint space, where motion dynamics are entangled with inter-agent positions, limiting scalability and requiring re-training when the number of agents changes. In contrast, HINT decouples individual motion representation from social interactions, while explicit relative transformations (rotations and translations) among agents are provided separately as conditions in diffusion. This separation enables the latent space to concentrate on motion semantics and ensures seamless adaptability to scenarios involving variable number of agents without finetuning or re-training. **Second**, we propose a sliding-window strategy for efficient online generation. Local to global temporal, spatial, and semantic cues are then aggregated to guide the diffusion process. Local conditions are collected inside each window, *i.e.*, target human's motion history, step index, partners' motion, and word-level text guidance, capturing fine-grained social and temporal dependencies within the window and preventing semantic drift. Global conditions, including sequence index of the current window, total frame length, and compositional command text guidance, are used to locate the current window within the entire sequence and thereby enforce long-term consistency. This hierarchical design enables natural, coherent, and semantically aligned multi-human motion generation, as shown in Fig. 1.

We conduct extensive experiments on the InterHuman (Liang et al., 2024) and InterX (Xu et al., 2024a) benchmarks. Results show that HINT not only matches the performance of strong offline methods but also outperforms existing autoregressive baselines by a large margin.

## 2 RELATED WORK

**Single Human Motion Generation.** Recent work approaches single-person motion generation mainly with diffusion or autoregressive models. Diffusion-based methods (Tevet et al., 2022; Chen et al., 2023; Zhang et al., 2024b; Barquero et al., 2024) capture complex distributions and yield high-quality sequences across modalities such as text, audio, and scene context (Tevet et al., 2022; Xu et al., 2023; Alexanderson et al., 2023), but are typically limited to fixed-length clips. Autoregressive models (Jiang et al., 2023; Zhang et al., 2023) instead generate motions step by step, enabling variable-length synthesis and finer control, though they are prone to error accumulation.

DART (Zhao et al., 2024) bridges these paradigms through a latent diffusion–autoregressive design that supports streaming, controllable motion generation. We build on this paradigm to extend autoregressive diffusion to multi-human interaction, explicitly modeling semantic dependencies and coordination between participants.

**Human-Human Interaction Generation.** Recent years have witnessed increasing interest in human interaction motion generation (Chopin et al., 2023; Xu et al., 2024b; Liu et al., 2024; Ghosh et al., 2024; Tan et al., 2025; Liang et al., 2024; Shafir et al., 2024; Ruiz-Ponce et al., 2024; Wang et al., 2024; Javed et al., 2024; Cai et al., 2024), particularly in the areas of reaction and interaction generation. Reaction generation aims to synthesize plausible responses conditioned on a partner's motion, with approaches ranging from Transformer-based coordination (Chopin et al., 2023) and diffusion with distance constraints (Xu et al., 2024b) to physics-driven modeling (Liu et al., 2024), spatio-temporal cross-attention (Ghosh et al., 2024), and reasoning with LLMs (Tan et al., 2025). Interaction generation instead models both humans jointly, using dual-branch diffusion (Liang et al., 2024), lightweight communication across pretrained models (Shafir et al., 2024), dual-level textual prompts (Ruiz-Ponce et al., 2024), LLM-based planning (Wang et al., 2024), or masked spatio-temporal token prediction (Javed et al., 2024). Beyond pairwise interactions, SocialGen (Yu et al., 2025) leverages language models for group social behaviors, Multi-Person Interaction Generation (Xu et al., 2025) scales two-person priors to larger groups, and PINO (Ota et al., 2025) enables long-duration and customizable generation for arbitrary group sizes. Despite these advances, most methods still generate fixed-length sequences, limiting their applicability in real-time and streaming scenarios.

## 3 METHOD

We address text-driven online multi-human motion generation, which sequentially predicts future poses of $N$ agents conditioned on their past motions and a textual description $\mathcal{T}$. Formally, let

$$\mathbf{M}^{1:T} = \left\{ \mathbf{m}_{(i)}^t \in \mathbb{R}^d \mid i = 1, \dots, N; \; t = 1, \dots, T \right\} \tag{1}$$

denote the motion sequence of $N$ humans over $T$ timesteps, where $\mathbf{m}_{(i)}^t$ is the motion representation of agent $i$ at time $t$, $d$ is the dimension of the representation. Autoregressive multi-human motion generation recursively predicts

$$\hat{\mathbf{M}}^{t:t+K} \sim p_\theta \left( \mathbf{M}^{t:t+K} \mid \hat{\mathbf{M}}^{1:t-1}, \mathcal{T}^{1:t+K} \right), \tag{2}$$

with trained parameters $\theta$, thereby capturing both temporal dependencies across timesteps and social dependencies across humans. We jointly predict $K$ future timesteps for efficiency. For clarity, we use $h_A^{1:H}$ to represent the $H$-timestep history motion of agent $A$ and $f_A^{1:K}$ to represent the $K$-timestep future motion within a sliding window, as shown in Fig. 3. We empirically set $H = 4, K = 16$.

### 3.1 OVERVIEW OF HINT

Fig. 3 demonstrates the **two-human** motion generation pipeline of **HINT**, which employs a sliding-window strategy to autoregressively extend future segments with a diffusion model (see Fig. 2). We show that HINT naturally generalizes to **multi-human** settings in Sec. 3.5

**Motion VAE.** In Fig. 3 (a), we first construct a canonicalized shared latent space to map raw motion sequences into latent representations. Concretely, the motion of each individual, $m_A, m_B \in \mathbb{R}^{T \times d}$, with $T$ timesteps and $d$ dimensions, is divided into overlapping windows and canonicalized in its local coordinate to remove absolute position. A transformer-based Motion VAE, following DART (Zhao et al., 2024), is then employed for sliding-window modeling. The VAE consists of an encoder $\mathcal{E}$ and a decoder $\mathcal{D}$. Given a window with $H$ history $h_{(i)}^{1:H}$ and $K$ future frames $f_{(i)}^{1:K}$ for human $i \in \{A, B\}$, $\mathcal{E}$ conditions on the history and encodes the future into a latent vector $\mathbf{z}_{(i)}^{\mathbf{f}} \in \mathbb{R}^l$ in the shared latent space, where $\mathbf{z}_{(i)}^{\mathbf{f}}$ denotes the future motion representation of agent $i$, and $l$ is the latent dimensionality. $\mathcal{D}$ reconstructs the $K$ future frames from $\mathbf{z}_{(i)}^{\mathbf{f}}$, conditioned on the corresponding history. Once trained, both encoder $\mathcal{E}$ and decoder $\mathcal{D}$ are frozen in subsequent modules, ensuring that the latent space remains stable and consistent.

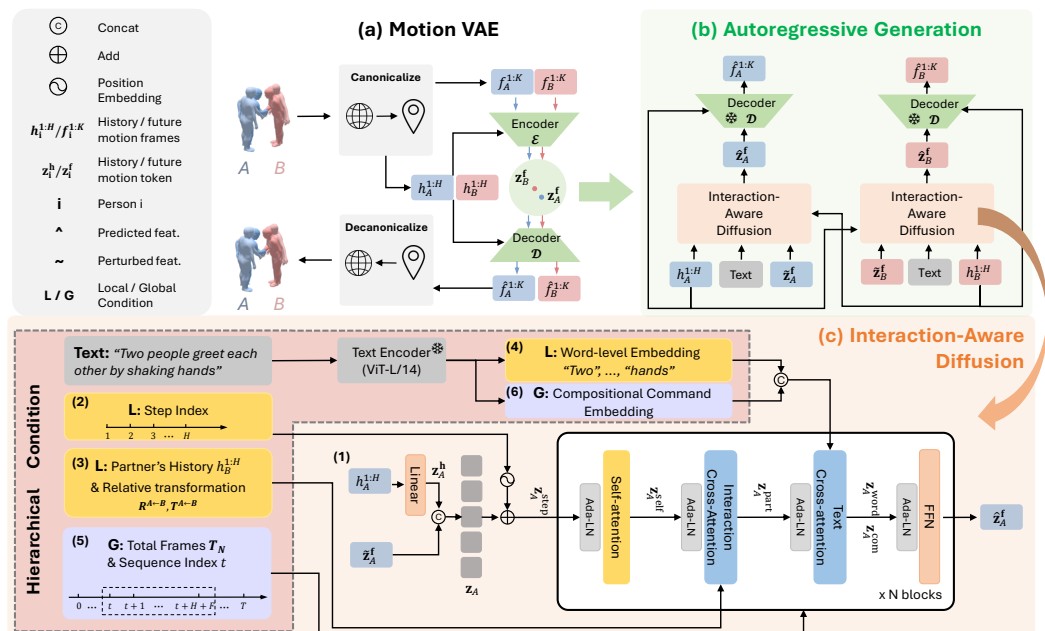

Figure 3: **Overview of HINT in two-human interaction generation.** (a) Canonicalized latent space. (b) Within this latent space, motion is generated in a sliding-window autoregressive manner, where the Interaction-Aware Diffusion predicts the next $K$ frames. (c) The detailed architecture of the Interaction-Aware Diffusion, in which hierarchical conditions guide the generation process.

**Sliding-Window Strategy for Autoregressive Generation.** As illustrated in Fig. 3 (b), HINT employs a sliding-window autoregressive process for both training and generation. During training, the ground-truth motion sequence is divided into overlapping windows. Future frames in each window are encoded into a latent $\mathbf{z}^{\mathbf{f}}_{(i)}, i \in \{A, B\}$ by the motion encoder $\mathcal{E}$, where noise is added to obtain $\tilde{\mathbf{z}}^{\mathbf{f}}_{(i)}$. Conditioned on $\tilde{\mathbf{z}}^{\mathbf{f}}_{(i)}$, historical motion $h^{1:H}_{(i)}$, and text $\mathcal{T}$, the Interaction-Aware Diffusion model learns to predict the denoised latent $\hat{\mathbf{z}}^{\mathbf{f}}_{(i)}$. During inference, we sample a future latent $\hat{\mathbf{z}}^{\mathbf{f}}_{(i)}$ conditioned on the history and text. Then the frozen decoder $\mathcal{D}$ reconstructs the future $K$ frames $\hat{f}^{1:K}_{(i)}$ from $\hat{\mathbf{z}}^{\mathbf{f}}_{(i)}$. $\hat{f}^{1:K}_{(i)}$ is appended to the history to condition the next window, proceeding autoregressively.

**Interaction-Aware Diffusion.** Fig. 3 (c) illustrates the Interaction-Aware Diffusion module, exemplified with two-human interaction generation. The model employs shared weights and shared conditioning signals for both agents $A$ and $B$, ensuring that the same model parameters are applied symmetrically across the two. Taking the motion generation of human $A$ as an example, the model is first conditioned on the historical motion sequences of both human $A$ and human $B$, denoted as $h^{1:H}_A$ and $h^{1:H}_B$, respectively. Then the relative rotation $\mathbf{R}^{A \leftarrow B}$ and translation $\mathbf{T}^{A \leftarrow B}$ are encoded and integrated, which are calculated from the history motion. Additionally, the module integrates a set of hierarchical conditions: 1) step index within the current temporal window; 2) word-level text embedding; 3) sequence index of the current window in the overall sequence, total frame length $T_N$ to be generated according to the textual description; and 4) compositional command embedding. Together, these conditions encode both textual semantics and multi-scale positional information, enabling the model to capture fine-grained temporal dependencies while maintaining global consistency across the generated motion sequence.

## 3.2 CANONICALIZED LATENT SPACE

We propose a *Canonicalized Latent Space*, as illustrated in Fig. 3 (a), which encodes explicit motion sequences into compact latents. Existing methods (Liang et al., 2024; Javed et al., 2024) on human-human interaction generation often adopt a Joint Multi-Human Latent Space by applying the same coordinate transformation to both agents' motions, which preserves relative position but entangles

motion semantics with inter-person geometry, hindering generalization to multi-human scenarios. Instead, we perform independent canonicalization and normalization for each agent $A$, $B$, transforming motions to their respective local coordinate, while explicitly encoding relative transformations (rotation $\mathbf{R}^{A \leftarrow B}$ and translation $\mathbf{T}^{A \leftarrow B}$) as conditions for the generative model. Specifically, after canonicalization, each individual is oriented toward the positive $z$-axis, with the root joint positioned at the origin.

Formally, given a motion sequence $\mathbf{M}_{(i)}$ for human $i \in \{A, B\}$, the canonicalized motion is defined as $\mathbf{M}^{\mathbf{c}}_{(i)} = \mathbf{R}_{(i)} \mathbf{M}_{(i)} + \mathbf{T}_{(i)}$, where $\mathbf{R}_{(i)}$ and $\mathbf{T}_{(i)}$ denote the rotation and the translation, respectively. Following the reparameterization strategy (Kingma & Welling, 2013), the latent representation $\mathbf{z}^{\mathbf{f}}_{(i)}$ is obtained via the encoder $\mathcal{E}$ as

$$\mathbf{z}^{\mathbf{f}}_{(i)} \sim q_\phi(\mathbf{z}^{\mathbf{f}}_{(i)} \mid \mathbf{M}^{\mathbf{c}}_{(i)}), \tag{3}$$

where $q_\phi$ is a Gaussian inference network. Training objectives are described in Sec. 3.4. To inject relative positional information, we compute the relative rigid transformation as follows,

$$\mathbf{R}^{i \leftarrow j} = \mathbf{R}_{(i)} \mathbf{R}^{\top}_{(j)}, \quad \mathbf{T}^{i \leftarrow j} = \mathbf{T}_{(i)} - \mathbf{R}^{i \leftarrow j} \mathbf{T}_{(j)}. \tag{4}$$

This transformation $[\mathbf{R}^{i \leftarrow j}, \mathbf{T}^{i \leftarrow j}]$ is encoded into the diffusion network as a condition term.

**Canonicalized Latent Space *vs.* Joint Multi-Human Latent Space**. Our canonicalized latent space has two advantages over previous joint multi-human latent space (Liang et al., 2024; Javed et al., 2024; Ruiz-Ponce et al., 2024). First, it effectively disentangles absolute position information from motion dynamics, forcing the latent to focus on the movement patterns themselves without being biased by spatial location. Second, such design enforces cross-human consistency in the latent space, thereby facilitating robust generalization to interactions involving three or more humans.

### 3.3 HIERARCHICAL MOTION CONDITION

To enable effective autoregressive motion generation, we incorporate local-to-global guidance into the diffusion process. Built upon a latent diffusion backbone, our *Hierarchical Motion Condition* (HMC) strategy organizes temporal, spatial, and semantic cues into multi-level conditions. We provide local conditions, which capture short-term dependencies and fine-grained semantic alignment within the current window, and global conditions, which enforce long-term consistency across the sequence described by the text guidance. We demonstrate HMC on human-human interaction generation (Fig. 3 (c)), illustrating the procedure from human $A$'s perspective, as weights and conditions are shared across both humans.

**Local Conditions.** Within each window, we employ four types of local conditions as follows.

*1) Target Human History Embedding.* As shown in Fig. 3 (c-1), the history motion of human $A$, $h^{1:H}_A$, is first mapped into a feature representation $\mathbf{z}^{\mathbf{h}}_A$ via a linear projection. $\mathbf{z}^{\mathbf{h}}_A$ is then concatenated with the future motion token $\mathbf{z}^{\mathbf{f}}_A$ to form the motion feature $\mathbf{z}_A$.

*2) Step Index.* Both human $A$ and $B$ provide $H$ history frames. Each history frame is indexed by its timestep from 1 to $H$, and the index is encoded into an embedding $\mathbf{e}_s$ (Fig. 3 (c-2)). Adding this to the motion feature $\mathbf{z}_A$ yields $\mathbf{z}^{\text{step}}_A = \mathbf{z}_A + \mathbf{e}_s$, which is then processed via self-attention to capture temporal dependencies:

$$\mathbf{z}^{\text{self}}_A = \text{SelfAttn}(\mathbf{z}^{\text{step}}_A, \mathbf{z}^{\text{step}}_A, \mathbf{z}^{\text{step}}_A). \tag{5}$$

This enables the model to reason about temporal ordering within the prediction window.

*3) Partner History Embedding.* To model interactions, human $B$'s history $h^{1:H}_B$ is transformed into human $A$'s local coordinate via relative rotation $\mathbf{R}^{A \leftarrow B}$ and translation $\mathbf{T}^{A \leftarrow B}$ (Fig. 3 (c-3)):

$$h^{1:H}_{B \rightarrow A} = \mathbf{R}^{A \leftarrow B} h^{1:H}_B + \mathbf{T}^{A \leftarrow B}. \tag{6}$$

This is integrated into the diffusion model through Interaction Cross-Attention:

$$\mathbf{z}^{\text{part}}_A = \text{CrossAttn}(\mathbf{z}^{\text{self}}_A, h^{1:H}_{B \rightarrow A}, h^{1:H}_{B \rightarrow A}). \tag{7}$$

*4) Word-Level Text Embedding.* Finally, we introduce word-level text embedding to impose fine-grained semantic fidelity within each window, as depicted in Fig. 3 (c-4). Since a single sentence

may be very long or contain complex commands, we split it into words, each serving as a token, $\mathbf{E}_{\text{word}} = [\mathbf{e}_1, \ldots, \mathbf{e}_L]$, where $\mathbf{e}_l$ denotes the embedding of the $l$-th token, and integrated into latent features via Text Cross-attention:

$$\mathbf{z}_A^{\text{word}} = \text{CrossAttn}(\mathbf{z}_A^{\text{part}}, \mathbf{E}_{\text{word}}, \mathbf{E}_{\text{word}}).$$

By jointly leveraging individual history, step index, partner history, and token-level text embedding, the model captures fine-grained interaction patterns and achieves precise text–motion alignment within each rollout, thereby alleviating semantic drift in long-sequence generation.

**Global Conditions.** Across all windows, we collect the following two types of global information.

*1) Sequence Index and Total Frame Number.* In Fig. 3 (c-5), we first incorporate both the global sequence index $t$ and the total number of frames $T_N$ of the corresponding text segment, which indicate the position of the current window within the entire motion sequence and the overall sequence length, respectively. This information is then injected into the diffusion network through Adaptive Layer Normalization (AdaLN) (Peebles & Xie, 2023), ensuring that the generation process is aware of both the frame-level position and the global temporal context.

During training and quantitative evaluation, we simply set $T_N$ to the ground-truth sequence length provided by the dataset, matching offline baselines that generate the entire sequence at once for a fair comparison. In deployment, however, $T_N$ can be flexibly specified or automatically selected according to practical needs (see Sec. B.4 for details).

*2) Compositional Command Embedding.* If the user provides a textual description for the entire sequence, where the description consists of multiple interconnected commands that drive the human body to achieve one or more specific goals, we encode the whole text $\mathcal{T}$ into a single global token $\mathbf{e}$ to serve as guidance for the sequence generation, as depicted in Fig. 3 (c-6). Then $\mathbf{e}$ is injected into the model through Text Cross-Attention to provide global semantic guidance across windows:

$$\mathbf{z}_A^{\text{com}} = \text{CrossAttn}(\mathbf{z}_A^{\text{part}}, \mathbf{e}, \mathbf{e}). \tag{8}$$

Conceptually, word-level embeddings $\mathbf{E}_{\text{word}} = [\mathbf{e}_1, \ldots, \mathbf{e}_L]$ serve as local conditions while compositional command embedding $\mathbf{e}$ functions as the global condition. In practice, however, both are concatenated and jointly fed into the same Text Cross-Attention block, allowing simultaneous modeling of fine-grained semantics and holistic context within a unified interaction.

### 3.4 TRAINING STRATEGY

We adopt a two-stage training strategy that decouples motion encoding and generation.

**Stage I: Motion VAE Pretraining.** The Motion VAE is pretrained to obtain stable latent representations by optimizing the standard VAE objective (Kingma & Welling, 2013):

$$\mathcal{L}_{\text{VAE}} = \sum_i \mathcal{L}_{\text{rec}}(\hat{\mathbf{M}}_{(i)}, \mathbf{M}_{(i)}) + \beta \, \mathcal{L}_{\text{KL}}\big(q_\phi(\mathbf{z}_{(i)}^{\mathbf{f}} \mid \mathbf{M}_{(i)}^{\mathbf{c}}) \,\|\, p(\mathbf{z}_{(i)}^{\mathbf{f}})\big), \tag{9}$$

where $\mathcal{L}_{\text{rec}}$ reconstructs future frames, $\mathcal{L}_{\text{KL}}$ regularizes the latent distribution with KL divergence, $\beta$ is a balancing factor, and $p(\mathbf{z}_{(i)}^{\mathbf{f}})$ denotes the standard Gaussian prior. After pretraining, the encoder and decoder are frozen in generation.

**Stage II: Diffusion with Autoregressive Sliding Window.** We train the Interaction-Aware Diffusion model using an autoregressive sliding-window strategy. At each window, the diffusion model is optimized with the standard denoising loss $\mathcal{L}_{\text{diff}}$, augmented by interaction-specific regularizers inspired by InterGen (Liang et al., 2024), including joint affinity $\mathcal{L}_{\text{aff}}$, cross-person distance constraint $\mathcal{L}_{\text{dist}}$, and relative orientation constraint $\mathcal{L}_{\text{ori}}$:

$$\mathcal{L} = \mathcal{L}_{\text{diff}} + \lambda_{\text{aff}}\mathcal{L}_{\text{aff}} + \lambda_{\text{dist}}\mathcal{L}_{\text{dist}} + \lambda_{\text{ori}}\mathcal{L}_{\text{ori}}, \tag{10}$$

where $\lambda_*$ indicates balancing weights. Please refer to Appendix B.2 for details of each loss term.

### 3.5 FROM TWO-HUMAN TO MULTI-HUMAN MOTION GENERATION

Built upon the Canonicalized Latent Space (Sec. 3.2) and the shared-weight Interaction-Aware Diffusion model (Fig. 3 (c)), HINT naturally generalizes to multi-human interaction scenarios. The

Table 1: Results on InterHuman and InterX. $\rightarrow$ denotes closer to ground truth is better, $\uparrow$ / $\downarrow$ means higher/lower is better, $\pm$ indicates the 95% confidence interval. **Bold** denotes the best result. InterMask* is the online version of InterMask, while DART$^{\dagger}$ is the two-human version of DART.

| Dataset | Setting | Method | R@Top3↑ | FID↓ | MM Dist↓ | Diversity→ |
|---------|---------|--------|---------|------|----------|-----------|
| Inter Human | | Ground Truth | $0.701^{\pm.008}$ | $0.273^{\pm.007}$ | $3.755^{\pm.008}$ | $7.948^{\pm.064}$ |
| | offline | T2M (Guo et al., 2022) | $0.464^{\pm.014}$ | $13.769^{\pm.072}$ | $5.731^{\pm.013}$ | $7.046^{\pm.022}$ |
| | | MDM (Tevet et al., 2022) | $0.339^{\pm.012}$ | $9.167^{\pm.056}$ | $7.125^{\pm.018}$ | $7.602^{\pm.045}$ |
| | | ComMDM (Shafir et al., 2024) | $0.466^{\pm.010}$ | $7.069^{\pm.054}$ | $6.212^{\pm.021}$ | $7.244^{\pm.038}$ |
| | | InterGen (Liang et al., 2024) | $0.624^{\pm.010}$ | $5.918^{\pm.079}$ | $5.108^{\pm.014}$ | $7.387^{\pm.029}$ |
| | | MoMat–MoGen (Cai et al., 2024) | $0.666^{\pm.004}$ | $5.674^{\pm.085}$ | $\mathbf{3.790}^{\pm.001}$ | $8.021^{\pm.350}$ |
| | | in2IN (Ruiz-Ponce et al., 2024) | $0.662^{\pm.009}$ | $5.535^{\pm.120}$ | $3.803^{\pm.002}$ | $7.953^{\pm.047}$ |
| | | **InterMask** (Javed et al., 2024) | $\mathbf{0.683}^{\pm.004}$ | $\mathbf{5.154}^{\pm.061}$ | $\mathbf{3.790}^{\pm.002}$ | $\mathbf{7.944}^{\pm.033}$ |
| | online | InterMask* | $0.557^{\pm.004}$ | $14.352^{\pm.133}$ | $3.852^{\pm.001}$ | $7.485^{\pm.032}$ |
| | | DART$^{\dagger}$ | $0.642^{\pm.005}$ | $4.979^{\pm.053}$ | $3.813^{\pm.001}$ | $7.950^{\pm.032}$ |
| | | **HINT** | $\mathbf{0.672}^{\pm.004}$ | $\mathbf{3.100}^{\pm.035}$ | $\mathbf{3.796}^{\pm.001}$ | $\mathbf{7.898}^{\pm.023}$ |
| InterX | | Ground Truth | $0.736^{\pm.003}$ | $0.002^{\pm.0002}$ | $3.536^{\pm.013}$ | $9.734^{\pm.078}$ |
| | offline | T2M (Guo et al., 2022) | $0.396^{\pm.005}$ | $5.481^{\pm.382}$ | $9.576^{\pm.006}$ | $2.771^{\pm.151}$ |
| | | MDM (Tevet et al., 2022) | $0.426^{\pm.005}$ | $23.701^{\pm.057}$ | $9.548^{\pm.014}$ | $5.856^{\pm.077}$ |
| | | ComMDM (Shafir et al., 2024) | $0.236^{\pm.004}$ | $29.266^{\pm.067}$ | $6.870^{\pm.017}$ | $4.734^{\pm.067}$ |
| | | InterGen (Liang et al., 2024) | $0.429^{\pm.005}$ | $5.207^{\pm.216}$ | $9.580^{\pm.011}$ | $7.788^{\pm.208}$ |
| | | **InterMask** (Javed et al., 2024) | $\mathbf{0.705}^{\pm.005}$ | $\mathbf{0.399}^{\pm.013}$ | $\mathbf{3.705}^{\pm.017}$ | $\mathbf{9.046}^{\pm.073}$ |
| | online | InterMask* | $0.169^{\pm.003}$ | $19.445^{\pm.199}$ | $7.885^{\pm.003}$ | $6.250^{\pm.007}$ |
| | | DART$^{\dagger}$ | $0.510^{\pm.003}$ | $8.600^{\pm.075}$ | $5.492^{\pm.014}$ | $8.405^{\pm.073}$ |
| | | **HINT** | $\mathbf{0.682}^{\pm.003}$ | $\mathbf{0.278}^{\pm.012}$ | $\mathbf{4.007}^{\pm.016}$ | $\mathbf{8.886}^{\pm.066}$ |

proposed space can be directly applied to multi-human motion generation without additional training, since it decouples individual motion with social interactions. For the diffusion model, we only update one condition term: partner history embedding (Sec. 3.3, Local Conditions) by directly concatenating all partners motion history and feeding them into the diffusion. We do not perform any fine-tuning when scaling to more humans. Here, we only provide the most straightforward extension from two-person to multi-person motion generation. If additional multi-person motion datasets are employed, fine-tuning the cross-attention module is expected to yield further performance improvements. Please refer to Supplementary Materials for video results.

## 4 EXPERIMENTS

**Datasets.** We evaluate HINT on InterHuman (Liang et al., 2024) and InterX (Xu et al., 2024a). *InterHuman* comprises 7,779 motion sequences paired with 23,337 unique textual annotations containing 5,656 distinct words. It is built upon the SMPL-H body model, and we adopt a motion representation similar to InterGen (Liang et al., 2024), where each frame is expressed as $x^i = [\mathbf{j}_l^p, \mathbf{j}_l^v, \mathbf{j}^r, \mathbf{c}^f]$. Here, $\mathbf{j}_l^p \in \mathbb{R}^{3N_j}$ and $\mathbf{j}_l^v \in \mathbb{R}^{3N_j}$ represent the joint positions and velocities in the normalized local frame, $\mathbf{j}^r \in \mathbb{R}^{6(N_j-1)}$ denotes the 6D rotation (Zhou et al., 2019) of each joint in the root frame, $\mathbf{c}^f \in \mathbb{R}^4$ is a binary foot-ground contact feature, and $N_j$ denotes the number of joints, set to 22 for InterHuman. *InterX* is based on the SMPL-X body model and contains 13,888 motion sequences with 34,164 fine-grained textual descriptions. Each motion frame is represented as $x^i = [\mathbf{j}^r, \mathbf{r}_l^p, \mathbf{r}_l^v]$, where $\mathbf{j}^r \in \mathbb{R}^{6N_j}$ is the 6D rotation in the normalized local frame, and $\mathbf{r}_l^p \in \mathbb{R}^3$, $\mathbf{r}_l^v \in \mathbb{R}^3$ denote the root joint's position and velocity in the local frame, respectively. For InterX, $N_j$ is set to 55.

**Baselines.** Offline baselines: T2M (Guo et al., 2022), MDM (Tevet et al., 2022), ComMDM (Shafir et al., 2024), InterGen (Liang et al., 2024), MoMat-MoGen (Cai et al., 2024), in2IN (Ruiz-Ponce et al., 2024), and InterMask (Javed et al., 2024) on InterHuman, and with T2M, MDM, ComMDM, InterGen, and InterMask on InterX. In addition, we introduce two extended baselines: InterMask*, which denotes our online adaptation of InterMask, and DART$^{\dagger}$, which denotes our extension of DART (Zhao et al., 2024) from single-human to two-human scenarios(see Appendix B.3 for details).

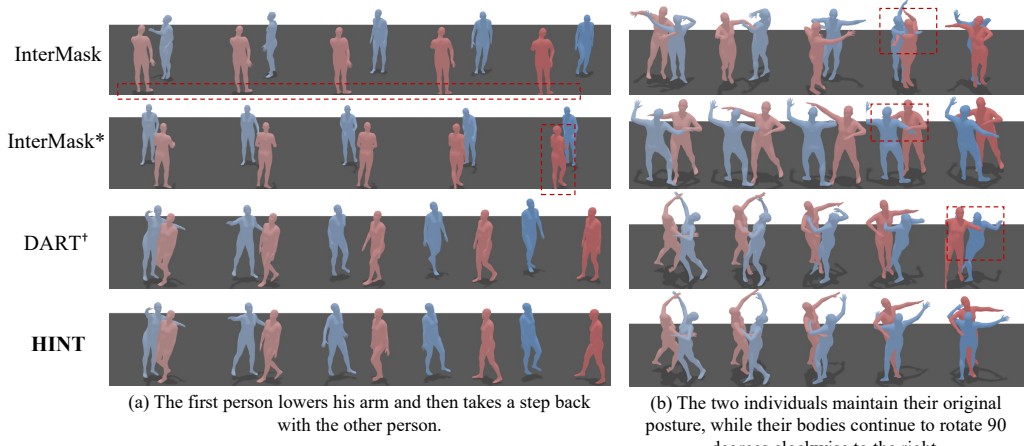

(a) The first person lowers his arm and then takes a step back with the other person.

(b) The two individuals maintain their original posture, while their bodies continue to rotate 90 degrees clockwise to the right.

Figure 5: Visual comparisons of InterMask, InterMask*, DART[†] and HINT on InterHuman. HINT performs better in regions with complex interactions.

**Evaluation Metrics.** *R-Precision* (reported as *R@Top3*; see Appendix C for *R@Top1/Top2*) and *Multimodal Distance* (*MM Dist*) are used to evaluate text-motion consistency. Specifically, *R-Precision* measures the rank of the Euclidean distance between motion and text embeddings, while *MM Dist* computes the average Euclidean distance between each generated motion and its corresponding text. *Frechet Inception Distance* (*FID*) evaluates the similarity in the feature space between generated and ground-truth motions, reflecting motion realism. *Diversity* (*Div*) measures motion variety via average pairwise feature distances among generated motions. All methods are evaluated 20 times with different random seeds, and we report the mean results with the $95\%$ confidence interval.

**Inference Speed.** HINT takes about 1.1s to generate 16 future frames from a single window on a single NVIDIA GeForce 3090 GPU, while DART[†] takes about 0.3s and InterMask* takes about 1.1s under the same conditions.

### 4.1 COMPARISON WITH BASELINES

Tab. 1 presents the evaluation results. Among all compared methods, HINT achieves state-of-the-art FID scores of 3.100 on InterHuman and 0.278 on InterX, improving 2.054 and 0.121 over the second-best method, InterMask. This significant gain highlights the superior realism and naturalness of the motions generated by HINT, which can be primarily attributed to HINT's hierarchical interaction modeling strategy. It explicitly and comprehensively conditions on past motion histories and relative position

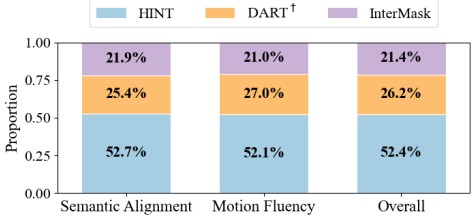

Figure 4: User study between HINT, DART[†] and InterMask.

relations between humans. As a result, within each sliding window, HINT is able to effectively capture and construct rich inter-human interactions. For other metrics, HINT is consistently superior to online competitors InterMask* and DART[†], while slightly inferior to the offline method InterMask. For instance, on InterHuman, compared to InterMask, HINT shows a small decrease of 0.011 in *R@Top3* and 0.006 in *MM Dist*. As an autoregressive method, HINT does not perform global optimization, which inevitably leads to insufficient alignment with the global text command. Overall, these experiment results validate HINT's effectiveness.

Fig. 5 shows qualitative comparisons of InterMask, InterMask*, DART[†], and HINT trained on InterHuman with the same text descriptions. In (a), InterMask fails to generate the backward motion of the left person, InterMask* produces less natural movements, while both DART[†] and HINT align well with the text. In (b), InterMask and InterMask* fail to generate motions consistent with the semantics, DART[†] does not explicitly model interactions and thus shows weaker interaction quality, whereas HINT achieves superior semantic alignment, interaction effectiveness, and motion fluency. More visual results are provided in Appendix C. Videos are provided in Supplementary Materials.

Table 2: Ablations HINT's key components on InterHuman. **L/G** indicates local/global conditions.

| | Method | R@Top3↑ | FID↓ | MM Dist↓ | Diversity→ |
|---|---|---|---|---|---|
| | Ground Truth | $0.701^{\pm.008}$ | $0.273^{\pm.007}$ | $3.755^{\pm.008}$ | $7.948^{\pm.064}$ |
| | w/o Canonicalized Latent Space | $0.633^{\pm.006}$ | $5.274^{\pm.051}$ | $3.814^{\pm.001}$ | $7.802^{\pm.025}$ |
| **L** | w/o History Motion Embedding | $0.660^{\pm.005}$ | $4.597^{\pm.063}$ | $3.802^{\pm.001}$ | $7.849^{\pm.030}$ |
| | w/o Step Index Embedding | $0.658^{\pm.005}$ | $3.224^{\pm.044}$ | $3.802^{\pm.001}$ | $7.875^{\pm.030}$ |
| | w/o Relative History Embedding | $0.647^{\pm.006}$ | $4.574^{\pm.058}$ | $3.808^{\pm.001}$ | $\mathbf{7.912}^{\pm.031}$ |
| | w/o Word-level Text Embedding | $\mathbf{0.672}^{\pm.004}$ | $3.295^{\pm.049}$ | $3.798^{\pm.001}$ | $7.908^{\pm.031}$ |
| **G** | w/o Sequence Index & Total Frame Number | $0.667^{\pm.004}$ | $3.543^{\pm.058}$ | $3.800^{\pm.001}$ | $7.874^{\pm.024}$ |
| | w/o Compositional Command Embedding | $0.669^{\pm.003}$ | $3.341^{\pm.045}$ | $3.797^{\pm.001}$ | $7.879^{\pm.024}$ |
| | **HINT** | $\mathbf{0.672}^{\pm.004}$ | $\mathbf{3.100}^{\pm.035}$ | $\mathbf{3.796}^{\pm.001}$ | $7.898^{\pm.023}$ |

**User Study.** To further evaluate the subjective quality of the generated results, we conducted a user study. Fifty participants are invited to compare HINT against InterMask and DART$^{\dagger}$ in terms of semantic alignment and motion fluency. Results are shown in Fig. 4. HINT received over $50\%$ of the votes across all metrics.

## 4.2 ABLATION STUDIES

Tab. 3 further compares our Canonicalized Latent Space (CLS) with the Joint Multi-Human Latent Space (JMLS) on the InterHuman dataset in terms of Reconstruction FID, MPJPE (Mean Per Joint Position Error), and MROE (Mean Relative Orientation Error), measuring reconstruction quality. For a fair comparison, we implement a motion VAE that di-

Table 3: Ablation of the Canonicalized Latent Space on InterHuman.

| Method | Recon FID↓ | MPJPE↓ | MROE↓ |
|---|---|---|---|
| Joint Multi-Human Latent Space | $7.783^{\pm.006}$ | $0.213^{\pm.001}$ | $0.426^{\pm.001}$ |
| **Canonicalized Latent Space** | $\mathbf{0.307}^{\pm.005}$ | $\mathbf{0.138}^{\pm.001}$ | $\mathbf{0.118}^{\pm.002}$ |

rectly encodes two-human motion trajectories as the JMLS baseline. The results demonstrate that CLS substantially outperforms JMLS in reconstruction quality (Recon FID: 0.307 *vs.* 7.783), highlighting that canonicalization enables more effective modeling of local human motion.

As shown in Tab. 2, we evaluate the contributions of HINT's key components, including the canonicalized latent space (CLS), local conditions (L), and global conditions (G). Replacing CLS with JMLS leads to a severe degradation in generation quality, with FID increasing from 3.100 to 5.274, underscoring its necessity. For local conditions, we remove individual components to assess their effectiveness. Excluding the history motion, step index, relative history, and word-level text embeddings results in slight R@Top3 drops of 0.012, 0.014, 0.025, and 0.000 (unchanged), respectively, compared to the full HINT (0.672). However, the corresponding FID values worsen significantly by 1.497, 0.124, 1.474, and 0.195. These consistent degradations verify that each local condition term provides complementary temporal or semantic cues and is indispensable for improving text-motion alignment and motion fidelity. For global conditions, the exclusion of compositional command embedding decreases R@Top3 by 0.003 and worsens FID by 0.241. Removing sequence index and total frame number has an even larger impact, with R@Top3 dropping by 0.005 and FID increasing by 0.443. These results highlight that both structural sequence information and compositional commands play crucial roles in ensuring coherent long-horizon motion generation and semantically grounded interaction synthesis.

## 5 CONCLUSION

In this paper, we presented HINT, the first autoregressive framework for multi-human motion generation with hierarchical interaction modeling in diffusion. By disentangling local motion semantics from inter-person interactions in a canonicalized latent space and adopting a sliding-window strategy that integrates both local and global context, HINT effectively adapts to varying numbers of human participants while maintaining long-horizon coherence. Extensive experiments on public

benchmarks demonstrate that HINT not only matches the performance of strong offline models but also significantly outperforms existing autoregressive baselines. In the future, an exciting direction is to extend our framework to incorporate objects and environments, enabling multi-human motion generation with object interactions. Text-driven generation of complex multi-agent behaviors in dynamic scenes remains a highly challenging yet impactful problem, and we believe HINT provides a strong foundation for advancing this line of research.

**Ethics Statement.** This work includes a user study to evaluate the perceptual quality of generated motion sequences. All participants were adult volunteers who provided informed consent prior to participation. No personally identifiable or sensitive information was collected. The study was conducted in accordance with standard academic ethical practices, and participants were free to withdraw at any time without consequence.

**Reproducibility Statement.** All experiments are conducted on publicly available datasets, and the implementation details, including model architectures, training procedures, and hyperparameters, are fully described in the main text and the appendix. Clear instructions for training and evaluation are provided in the Supplementary Material to ensure that the reported results can be reproduced under the same settings. In addition, we will release the source code and pretrained models to facilitate verification and further research by the community.

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

## A EXTENSION TO MULTI-HUMAN MOTION GENERATION

Our method can be naturally extended to multi-human interaction scenarios. Building upon the two-person generation framework, we simply incorporate the motion histories of additional participants into the conditioning to achieve joint modeling of multiple agents without modifying the architecture. In the two-person setting, all conditioning terms are defined relative to the current target agent, and thus remain valid when scaling to more agents. The only term that needs adaptation is the partner-history condition: for two agents, we pad the partner's motion history to a fixed length; for more than two agents, we concatenate the histories of all partners and then apply zero padding. To improve robustness, the location of the partner-history segment is randomly shifted during training.

Although the model is trained solely on two-person datasets, we observe that this strategy generalizes surprisingly well to multi-human interactions (as illustrated in Fig. A-1).

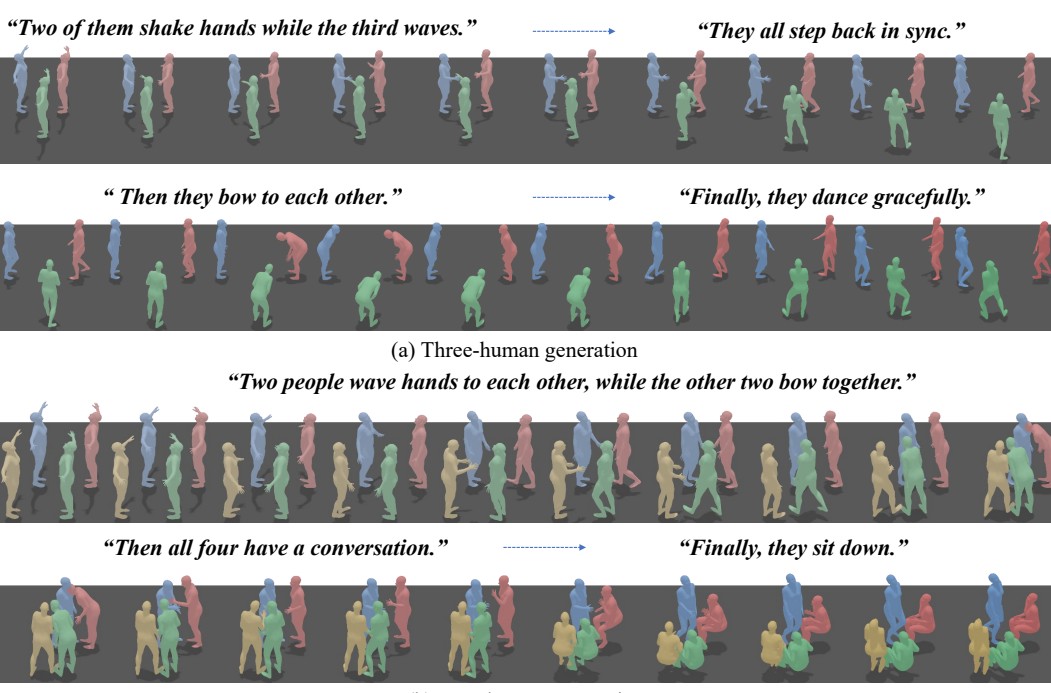

*"Two of them shake hands while the third waves."* --------→ *"They all step back in sync."*

*" Then they bow to each other."* --------→ *"Finally, they dance gracefully."*

(a) Three-human generation

*"Two people wave hands to each other, while the other two bow together."*

*"Then all four have a conversation."* --------→ *"Finally, they sit down."*

(b) Four-human generation

Figure A-1: Additional examples of three-human and four-human motion generation result.

We use HOI-M$^{3*}$ to further illustrate the effectiveness of HINT. The HOI-M$^{3*}$ dataset is extended from HOI-M$^3$ (Zhang et al., 2024a). HOI-M$^{3*}$ contains 52 videos (each approximately 6 minutes long) with 1,919 corresponding atomic textual descriptions. Fig. A-2 shows the configuration of HOI-M$^{3*}$. Quantitative evaluation results on HOI-M$^{3*}$ are demonstrated in Tab. A-1, while qualitative results are shown in Fig. A-3. More videos are provided in the supplementary material.

Table A-1: Results on HOI-M$^{3*}$. → denotes closer to ground truth is better, ↑ / ↓ means higher/lower is better, ± indicates the 95% confidence interval.

| Method | Segment | | | | Transition | | | |
|---|---|---|---|---|---|---|---|---|
| | R@Top3↑ | FID↓ | MM Dist↓ | Diversity→ | FID↓ | Diversity→ | PerkJerk→ | AUJ↓ |
| Ground Truth | $0.881^{\pm.002}$ | $0.001^{\pm.003}$ | $7.176^{\pm.001}$ | $6.809^{\pm.034}$ | $0.001^{\pm.000}$ | $6.113^{\pm.028}$ | $0.164^{\pm.001}$ | $0.005^{\pm.001}$ |
| HINT | $0.460^{\pm.004}$ | $1.501^{\pm.009}$ | $8.202^{\pm.004}$ | $6.849^{\pm.042}$ | $1.305^{\pm.021}$ | $6.188^{\pm.026}$ | $0.442^{\pm.005}$ | $0.272^{\pm.005}$ |

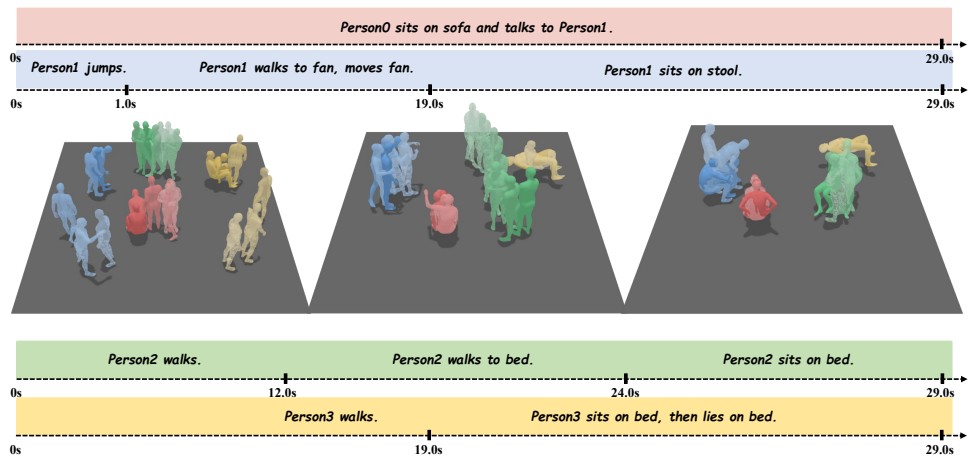

Figure A-2: Configuration of HOI-M$^{3*}$.

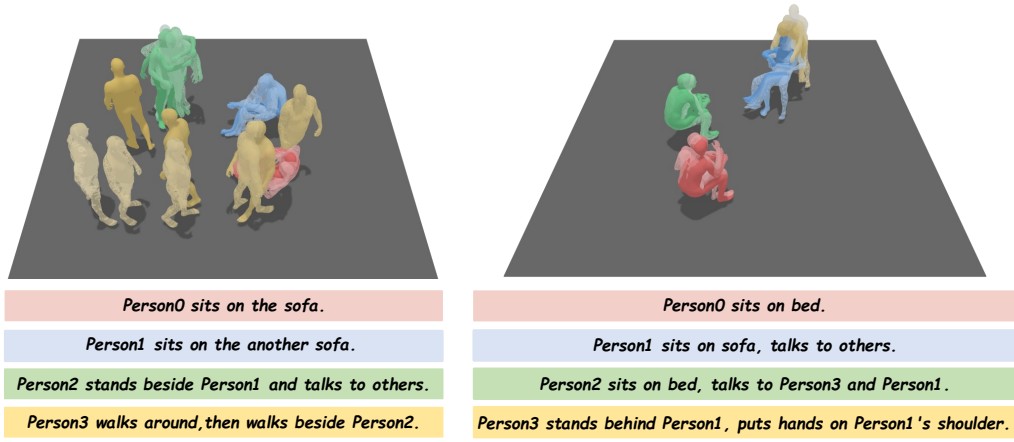

Figure A-3: Visualization Results on HOI-M$^{3*}$.

## B IMPLEMENTATION DETAILS

We provide more details of the model architecture, training, and the compared baselines. The implementation code of HINT is provided in the supplementary material as an attachment.

### B.1 MODEL ARCHITECTURE

**Motion VAE.** The Motion VAE adopts a transformer-based encoder–decoder architecture. Both encoder and decoder are constructed from stacked Transformer layers with residual connections and learned positional encodings. Raw motion sequences (history and future) are first linearly projected into a hidden space, and concatenated with a set of learnable global motion tokens. The encoder outputs the mean and variance of a Gaussian distribution, from which latent variables are sampled using the reparameterization trick. For the decoder, two variants are supported: we use all-encoder structure, where latent vectors and history embeddings are concatenated with query tokens and passed through a symmetric Transformer encoder.

**Interaction-Aware Diffusion.** The denoiser consists of $L_{diff} = 8$ transformer blocks with $H = 4$ heads, hidden size $d = 512$, and feed-forward width $d_{\text{ff}} = 1024$. It incorporates our Hierarchical Motion Condition (HMC), which fuses **local** conditions (individual history, step index, token-level text embedding, partners' history), **global** conditions (sequence length, sentence-level text embed-

ding), through self- and cross-attention, enabling both fine-grained alignment and global consistency.

Tab. B-2 shows the details model parameters of Motion VAE and Interaction-Aware Diffusion.

Table B-2: Parameters of Motion VAE and Interaction-Aware Diffusion.

| Parameter | Value |
|---|---|
| Latent dim ($d_z$) | 256 |
| Hidden dim ($d_h$) | 512 |
| Feed-forward dim ($d_{\text{ff}}$) | 1024 |
| Layers for VAE ($L_{VAE}$) | 5 |
| Transformer blocks of diffusion ($L_{diff}$) | 8 |
| Attention heads ($H$) | 4 |
| Dropout ($p$) | 0.1 |
| CLIP version | ViT-L/14@336px |

Table B-3: Training hyperparameters for Motion VAE and Interaction-Aware Diffusion.

| Hyperparameter | Value |
|---|---|
| stage1_steps | 100,000 |
| stage2_steps | 100,000 |
| stage3_steps | 100,000 |
| learning_rate | $10^{-4}$ |
| $\beta$ | $10^{-4}$ |
| $\lambda_{\text{aff}}$ | $10^{-1}$ |
| $\lambda_{\text{dist}}$ | $10^{-1}$ |
| $\lambda_{\text{ori}}$ | $10^{-4}$ |
| $\overline{D}_1$ | $10^{-1}$ |
| $\overline{D}_2$ | 1.0 |

## B.2 TRAINING DETAILS

**Three-stage Training Strategy for Motion VAE.** We adopt a three-stage training strategy for the Motion VAE and Interaction-Aware Diffusion following DART (Zhao et al., 2024):

*Stage I (Ground-Truth History)*: The model is trained on motion windows fully extracted from ground-truth sequences.

*Stage II (Mixed History)*: We gradually introduce predicted windows as part of the historical context. Specifically, the probability of replacing ground-truth history with the model's predictions is linearly increased during training.

*Stage III (Predicted History)*: The model is trained with history composed entirely of its own predicted windows, ensuring robustness in fully autoregressive generation.

**Detailed Loss Definition for Interaction-Aware Diffusion.** Following InterGen (Liang et al., 2024), the detailed definition of regularization loss is as follows:

*1) Joint affinity*

$$\mathcal{L}_{\text{aff}} = \left\| \left( D(m_A, m_B) - D(\hat{m}_A, \hat{m}_B) \right) \odot \mathbf{I}\left( D(m_A, m_B) < \overline{D}_1 \right) \right\|_2^2, \tag{B-1}$$

where $D(\cdot)$ computes the pairwise joint distance matrix, $\mathbf{I}(\cdot)$ is the indicator function, and $\overline{D}_1$ denotes a predefined distance threshold. This loss encourages the predicted motion $(\hat{m}_A, \hat{m}_B)$ to preserve the joint-level spatial affinity observed in the ground truth $(m_A, m_B)$.

*2) Distance map*

$$\mathcal{L}_{\text{dist}} = \left\| \left( D(m_A, m_B) - D(\hat{m}_A, \hat{m}_B) \right) \odot \mathbf{I}\left( D(\hat{m}_A, \hat{m}_B) < \overline{D}_2 \right) \right\|_2^2, \tag{B-2}$$

$\mathcal{L}_{\text{dist}}$ enforces accurate modeling of close-range spatial relationships while ignoring distant pairs that are less critical for interaction.

*3) Relative orientation*

$$\mathcal{L}_{\text{ori}} = \left\| O(m_A, m_B) - O(\hat{m}_A, \hat{m}_B) \right\|_2^2, \tag{B-3}$$

where $O(\cdot_A, \cdot_B)$ denotes the 6D representation (Zhou et al., 2019) of the relative rotation matrix from human A to B. $\mathcal{L}_{\text{ori}}$ enforces the predicted motions to preserve the relative orientations between the two humans, ensuring coherent and physically plausible interactions.

We also use a truncated regularization strategy: the regularization loss is only applied at lower diffusion timesteps. This prevents the denoiser from being biased towards implausible averaged poses and ensures more realistic motion generation.

Tab. B-3 presents the key hyperparameters of Motion VAE and Interaction-Aware Diffusion.

### B.3 DETAILS OF COMPARED METHODS

The details of baselines are as follows:

**Offline Methods.** T2M (Guo et al., 2022) is a Transformer-based motion generation framework that formulates text-conditioned motion synthesis as a sequence-to-sequence problem in a learned motion latent space. MDM (Tevet et al., 2022) employs a diffusion-based approach that conditions the denoising process on text or other modalities to generate high-quality, temporally coherent motions. ComMDM (Shafir et al., 2024) adds a lightweight communication block between two frozen MDM pretrained models to enable few-shot human-human interaction generation. InterGen (Liang et al., 2024) incorporates a mutual attention mechanism into the diffusion process to explicitly model inter-person dependencies for multi-person interaction generation. MoMat-MoGen (Cai et al., 2024) combines motion matching with generative modeling to enhance diversity while preserving motion naturalness, enabling high-quality text-driven motion synthesis. in2IN (Ruiz-Ponce et al., 2024) leverages both individual motion descriptions and global interaction semantics to improve diversity and accuracy in human–human interaction generation. InterMask (Javed et al., 2024) encodes motions as 2D token maps and jointly predicts masked tokens for both characters, enabling high-fidelity and diverse interaction generation. We use the results reported in their original papers.

**Online Extensions.** For InterMask*, we retain the 2D VQ-VAE structure of the original InterMask (Javed et al., 2024), but retrain it under our sliding-window setting to adapt to online generation. On top of this representation, we employ our autoregressive framework: within each prediction window, InterMask is used for motion generation. During generation, the history morion remains unmasked. And for DART$^{\dagger}$, we retrain DART (Zhao et al., 2024) on the InterHuman (Liang et al., 2024) and InterX (Xu et al., 2024a) dataset, where both humans share the same network during generation.

### B.4 EXPERIMENTS SETUP

#### B.4.1 HOW TO SET TOTAL FRAMES $T_N$?

$T_N$ is directly provided by the dataset. Similar to offline generation methods, which generate the entire sequence using the ground-truth length, we also use the ground-truth sequence length as the total generation length for a fair comparison.

However, in real applications, $T_N$ can be determined in several practical ways,

1. User-controlled based on instruction complexity. Empirically, users can assign a suitable $T_N$ according to the complexity of the textual command.

2. Using a large $T_N$ and trimming afterward. One may set $T_N$ to a sufficiently large value and then manually trim the generated video once the intended action has been completed.

3. Automatic stopping with a language–motion similarity metric (Our Eval Model). A more advanced option is to set $T_N$ to a relatively large upper bound and use the evaluation model we employ for computing MM-Dist. This model measures the similarity between the generated motion and the given textual description, and generation can stop as soon as the similarity surpasses a predefined threshold.

#### B.4.2 HOW TO DETERMINE HISTORY MOTION?

During training and quantitative evaluation, we simply use the first few frames of the ground-truth sequence as the history motion. At inference time, our system supports two types of initialization:

1. User-provided motion history. In scenarios that require precise control or integration with external systems, the user can provide a short initial sequence for each human. The model then performs online generation conditioned on this history.

2. Initialization from scratch under text-only conditioning. When only a textual description is given and no external history is available, we adopt a simple, unified initial pose such as a standing pose or

Table C-4: Detailed R-precision results on InterHuman and InterX. **Bold** denotes the best result for each setting.

| Dataset | Setting | Method | R-Precision↑ | | |
|---|---|---|---|---|---|
| | | | Top 1 | Top 2 | Top 3 |
| Inter Human | | Ground Truth | $0.452^{\pm.008}$ | $0.610^{\pm.009}$ | $0.701^{\pm.008}$ |
| | offline | T2M (Guo et al., 2022) | $0.238^{\pm.012}$ | $0.325^{\pm.010}$ | $0.464^{\pm.014}$ |
| | | MDM (Tevet et al., 2022) | $0.153^{\pm.012}$ | $0.260^{\pm.009}$ | $0.339^{\pm.012}$ |
| | | ComMDM (Shafir et al., 2024) | $0.223^{\pm.009}$ | $0.334^{\pm.008}$ | $0.466^{\pm.010}$ |
| | | InterGen (Liang et al., 2024) | $0.371^{\pm.010}$ | $0.515^{\pm.012}$ | $0.624^{\pm.010}$ |
| | | MoMat–MoGen (Cai et al., 2024) | $\mathbf{0.449}^{\pm.004}$ | $0.591^{\pm.003}$ | $0.666^{\pm.004}$ |
| | | in2IN (Ruiz-Ponce et al., 2024) | $0.425^{\pm.008}$ | $0.576^{\pm.008}$ | $0.662^{\pm.009}$ |
| | | **InterMask** (Javed et al., 2024) | $\mathbf{0.449}^{\pm.004}$ | $\mathbf{0.599}^{\pm.005}$ | $\mathbf{0.683}^{\pm.004}$ |
| | online | InterMask* | $0.331^{\pm.005}$ | $0.471^{\pm.005}$ | $0.557^{\pm.004}$ |
| | | DART$^{\dagger}$ | $0.395^{\pm.005}$ | $0.553^{\pm.005}$ | $0.642^{\pm.005}$ |
| | | **HINT** | $\mathbf{0.432}^{\pm.004}$ | $\mathbf{0.587}^{\pm.004}$ | $\mathbf{0.672}^{\pm.004}$ |
| InterX | | Ground Truth | $0.429^{\pm.004}$ | $0.626^{\pm.003}$ | $0.736^{\pm.003}$ |
| | offline | T2M (Guo et al., 2022) | $0.184^{\pm.010}$ | $0.298^{\pm.006}$ | $0.396^{\pm.005}$ |
| | | MDM (Tevet et al., 2022) | $0.203^{\pm.009}$ | $0.329^{\pm.007}$ | $0.426^{\pm.005}$ |
| | | ComMDM (Shafir et al., 2024) | $0.090^{\pm.002}$ | $0.165^{\pm.004}$ | $0.236^{\pm.004}$ |
| | | InterGen (Liang et al., 2024) | $0.207^{\pm.004}$ | $0.335^{\pm.005}$ | $0.429^{\pm.005}$ |
| | | **InterMask** (Javed et al., 2024) | $\mathbf{0.403}^{\pm.005}$ | $\mathbf{0.595}^{\pm.004}$ | $\mathbf{0.705}^{\pm.005}$ |
| | online | InterMask* | $0.061^{\pm.004}$ | $0.119^{\pm.003}$ | $0.169^{\pm.003}$ |
| | | DART$^{\dagger}$ | $0.252^{\pm.003}$ | $0.402^{\pm.003}$ | $0.510^{\pm.003}$ |
| | | **HINT** | $\mathbf{0.386}^{\pm.005}$ | $\mathbf{0.572}^{\pm.004}$ | $\mathbf{0.682}^{\pm.003}$ |

T-pose. The diffusion model then rolls out the full motion sequence from this starting state, guided by the text and interaction design.

### B.4.3 HOW TO CONTROL ERROR ACCUMULATION ACROSS WINDOWS?

Error accumulation across windows can be controlled using the following methods,

1. Short-window latent prediction reduces error propagation. Rather than rolling out frame by frame, HINT operates in a canonicalized latent space and predicts future motion in short windows (history length $H$, future length $K$). Each step only propagates errors at the window level, and the diffusion process refines a coherent latent trajectory within each window, which empirically stabilizes long-horizon rollouts.

2. Hierarchical conditions and canonicalized latent space prevent drift. Local conditions enforce short-term physical and social consistency inside each window, while global conditions anchor each window to the overall script, preventing long-term semantic drift. In addition, encoding motion in canonicalized per-person coordinates while feeding global geometry as explicit relative transforms decouples global position from motion semantics, reducing the amplification of small pose errors over time.

3. Autoregressive generation enables interactive correction. Since our model is online, users may issue light steering commands (e.g., "move slightly to the right") to correct deviations, a capability not available in offline methods. This is an inherent advantage of an online autoregressive design.

4. Global goal guided autoregressive generation. If the dataset provides additional global anchors, for example, a text prompt such as "go to the bed and sit on the bed", then the location of the bed can be supplied in advance as a conditioning term to the diffusion network. This global goal serves as a high-level anchor that guides the local window predictions, ensuring that the generated motion does not drift away from the intended final objective.

Table C-5: Detailed R-precision results of ablation studies on InterHuman. **L** and **G** indicate local and global conditions, respectively.

| Method | | R Precision↑ | | |
| --- | --- | --- | --- | --- |
| | | R@Top1 | R@Top2 | R@Top3 |
| Ground Truth | | $0.452^{\pm.008}$ | $0.610^{\pm.009}$ | $0.701^{\pm.008}$ |
| w/o Canonicalized Latent Space | | $0.396^{\pm.005}$ | $0.548^{\pm.005}$ | $0.633^{\pm.006}$ |
| **L** | w/o History Motion Embedding | $0.421^{\pm.006}$ | $0.576^{\pm.005}$ | $0.660^{\pm.005}$ |
| | w/o Step Index Embedding | $0.413^{\pm.005}$ | $0.570^{\pm.007}$ | $0.658^{\pm.005}$ |
| | w/o Relative History Embedding | $0.405^{\pm.005}$ | $0.563^{\pm.005}$ | $0.647^{\pm.006}$ |
| | w/o Word-level Text Embedding | $0.429^{\pm.006}$ | $\mathbf{0.591^{\pm.006}}$ | $\mathbf{0.672^{\pm.004}}$ |
| **G** | w/o Sequence Index & Total Frames Embedding | $0.425^{\pm.004}$ | $0.584^{\pm.004}$ | $0.667^{\pm.004}$ |
| | w/o Compositional Command Embedding | $0.420^{\pm.005}$ | $0.582^{\pm.003}$ | $0.669^{\pm.003}$ |
| **HINT** | | $\mathbf{0.432^{\pm.004}}$ | $0.587^{\pm.004}$ | $\mathbf{0.672^{\pm.004}}$ |

## C ADDITIONAL VISUALIZATION RESULTS

**Quantitative Results.** Detailed R-Precision results for InterHuman and InterX are presented in Tab. C-4. The ablation R-Precision results are reported in Tab. C-5.

**Qualitative Results.** More visualization results are shown in Figs. C-4–C-9.

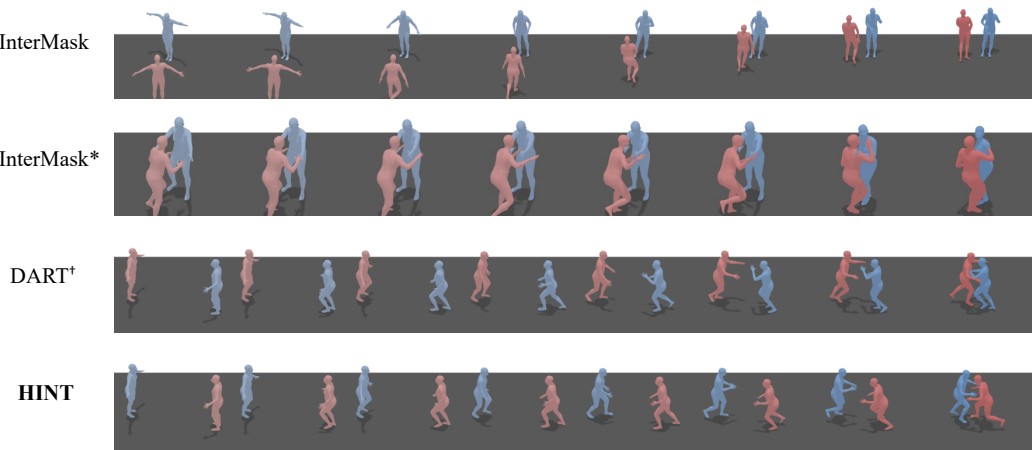

Figure C-4: They rush towards each other.

## D LIMITATION

In our experiments on the InterHuman dataset, we adopt a joint-based representation to ensure fair comparison with prior methods. For visualization, the corresponding SMPL parameters are reconstructed via inverse kinematics. Similarly, on the InterX dataset, we also restrict the representation to joint rotations. As a result, body penetration may occur. This limitation could be alleviated by enriching the representation space, introducing mesh-aware loss functions, and incorporating guidance during sampling.

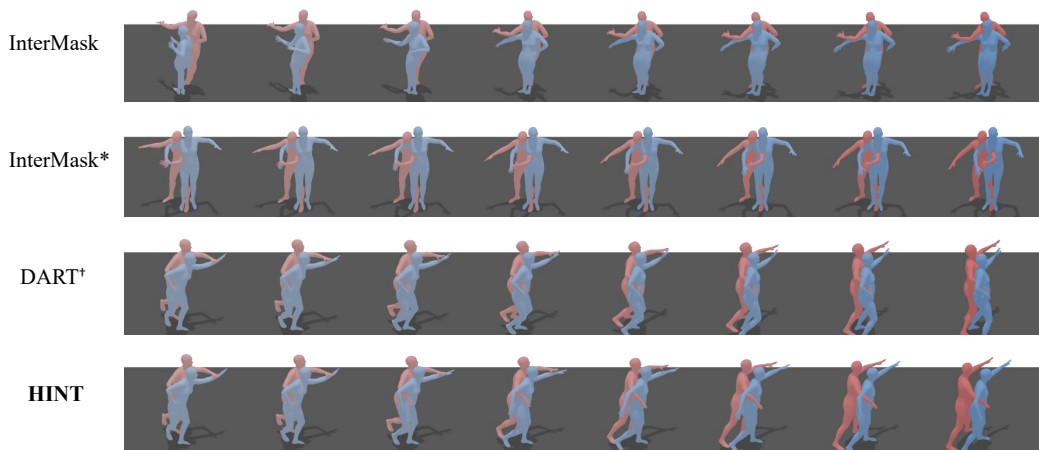

Figure C-5: The two people take a small step to the right side with their right foot.

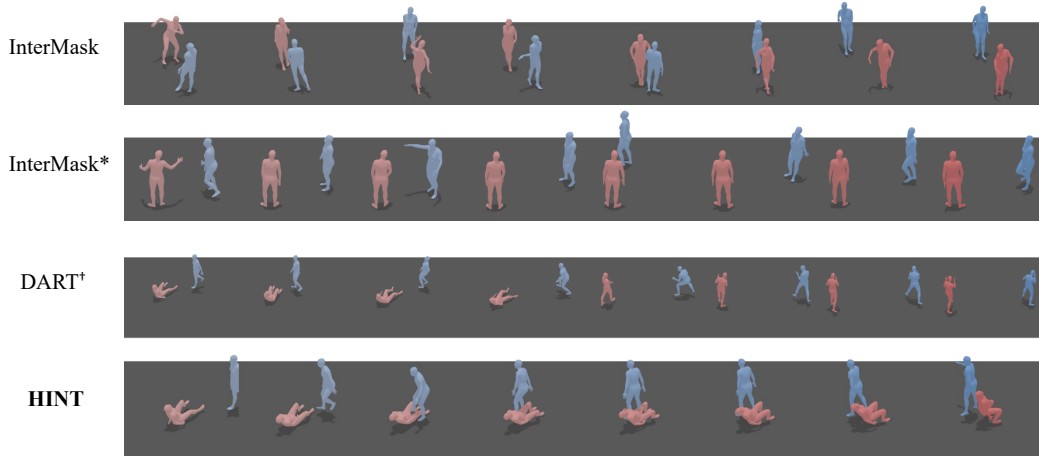

Figure C-6: The first one walks towards the second one and takes a few steps, pretends to hit the second one with the right hand, and then walks away to the side.

## E    DETAILS OF USER STUDY

We randomly sampled 30 textual descriptions from the test set of InterHuman (Liang et al., 2024). For each description, motion videos of the same length as the ground truth are generated using HINT, DART†, and InterMask (Javed et al., 2024). An online questionnaire is then distributed, where participants viewed the text and the corresponding videos and selected the best video based on semantic alignment and motion fluency. In total, 15 participants completed the survey. Fig. E-10 shows a screenshot of the questionnaire interface.

## F    USE OF LLMS

This paper used large language models (LLMs) to assist with language polishing. No core ideas, analyses, or experimental results were generated by LLMs.

## G    PHYSICAL REALISM, INTER-BODY CONTACT, AND PENETRATION

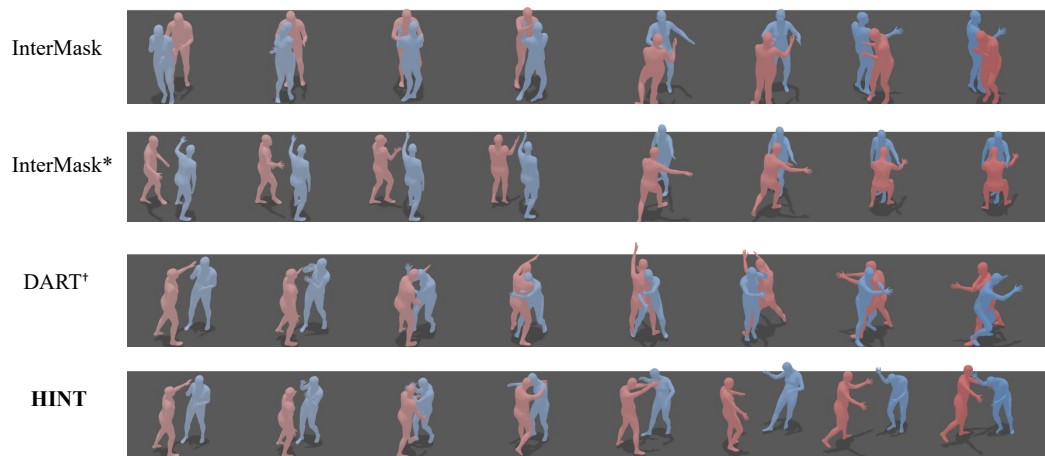

Figure C-7: One person extends their left arm and pushes the other person's right arm, and then they push each other back and forth.

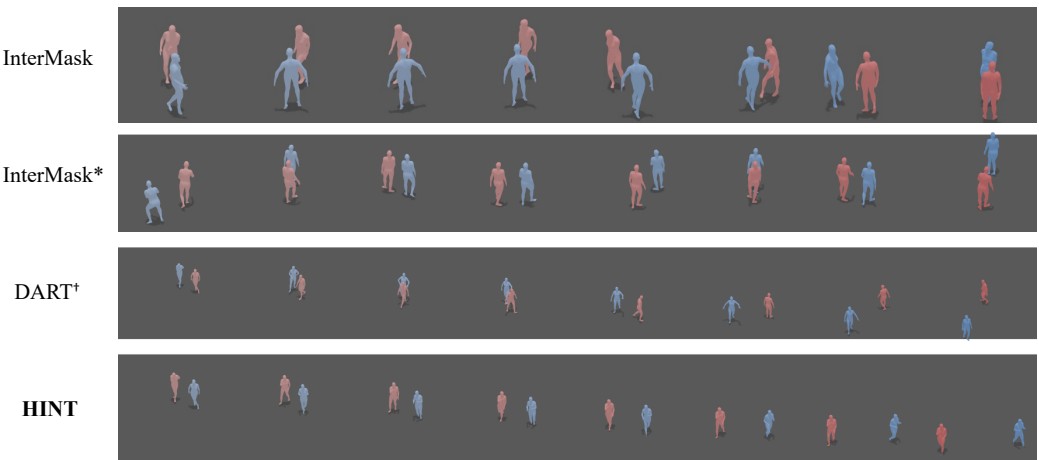

Figure C-8: Two people proceed ahead together.

Physical plausibility is an important objective in multi-human motion generation. In HINT, the main mechanism for encouraging realistic interactions is architectural rather than constraint-based: we explicitly encode the relative transformations between humans (e.g., relative rotations and translations) and feed this information as conditions to the diffusion network. This design allows the model to learn multi-person interaction patterns and contact behaviors directly from data.

For completeness, we further investigate an optional physics-aware loss that explicitly penalizes inter-penetration between human bodies. Instead of constructing a dense volumetric SDF for the full mesh, we adopt a lightweight *skeletal SDF* approximation: each human body is represented as a union of overlapping spheres centered at major joints or along limb segments. This choice is both computationally efficient and well aligned with the underlying kinematic structure of our SMPL(-X) skeleton.

Concretely, let $\mathcal{M}_B$ denote the body of human $B$ in a given frame, and let $\{(\mathbf{c}_j, r_j)\}_{j=1}^{J}$ be the centers and radii of the $J$ spheres used to approximate $\mathcal{M}_B$ (typically attached to joints or bones). We define the signed distance field of $B$ at a query point $\mathbf{x} \in \mathbb{R}^3$ as

$$d_B(\mathbf{x}) = \min_{1 \le j \le J} \left( \|\mathbf{x} - \mathbf{c}_j\|_2 - r_j \right) \tag{G-4}$$

where $d_B(\mathbf{x}) < 0$ indicates that $\mathbf{x}$ lies inside at least one sphere (i.e., inside the approximated body volume), $d_B(\mathbf{x}) = 0$ corresponds to the surface, and $d_B(\mathbf{x}) > 0$ is outside. This analytic,

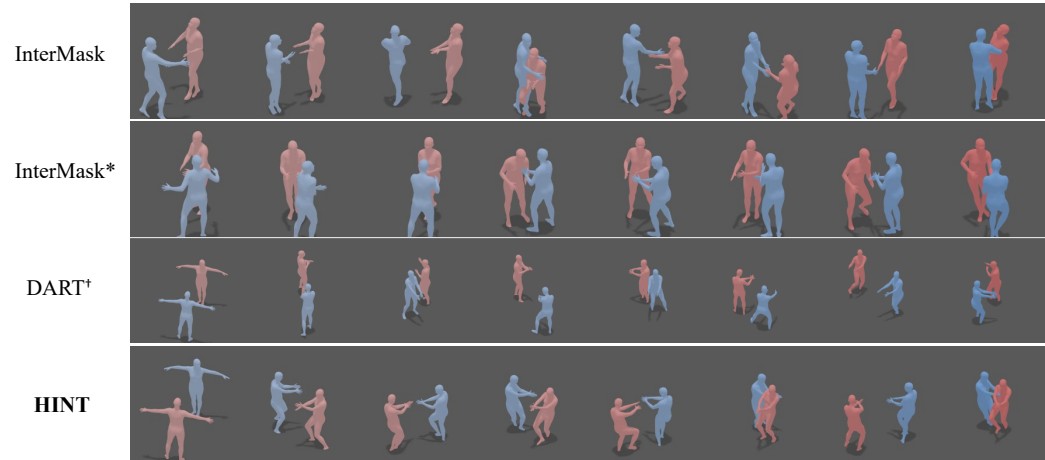

InterMask

InterMask*

DART†

**HINT**

Figure C-9: Both they encircle joining hands.

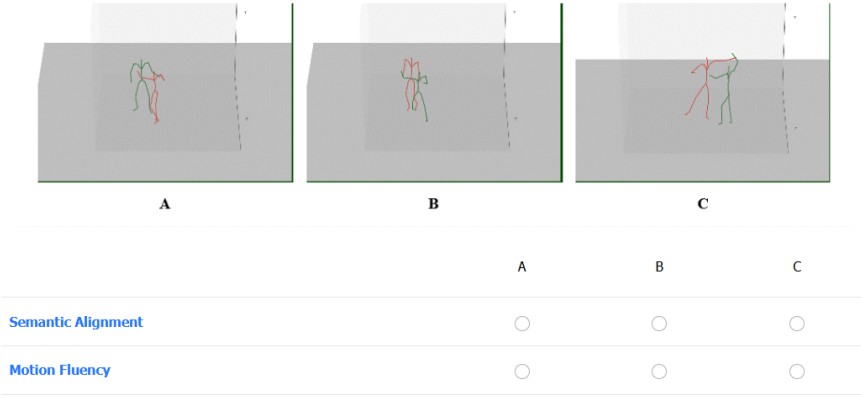

## Motion Generation: A User Study

Thank you for participating in this survey.
This study aims to compare the subjective performance of different motion generation methods for academic research purposes, and does not involve any privacy issues.
The questionnaire consists of 30 questions, each presenting three generated videos. For each question, please select the result that best matches the textual description and the result with the most fluent motion. The estimated completion time is 20–30 minutes.

* **01** Text: One person steps his left foot to the right and walks to the right, while the other person holds onto one person's right hand, and one person swings his left arm away to the right side.

```
1. One person steps his left foot to the right and walks to the right, while the other
person holds onto one person's right hand, and one person swings his left arm away to the
right side.
```

A    B    C

| | A | B | C |
|---|---|---|---|
| Semantic Alignment | ○ | ○ | ○ |
| Motion Fluency | ○ | ○ | ○ |

Figure E-10: Interface of the User Study.

joint-centered SDF behaves similarly to a mesh-based SDF near the body surface, while avoiding expensive mesh–mesh distance queries.

We then sample a set of points $\mathcal{P}_A$ from human $A$ (e.g., the mesh vertices) and define the penetration loss from $A$ into $B$ as

$$\mathcal{L}_{\text{pen}}(A \to B) = \frac{1}{|\mathcal{P}_A|} \sum_{\mathbf{p} \in \mathcal{P}_A} \big(\max(0, -d_B(\mathbf{p}))\big)^2 \tag{G-5}$$

Only points with $d_B(\mathbf{p}) < 0$ (i.e., inside $B$'s spherical proxy) contribute to the loss; points outside yield zero penalty. The total inter-penetration loss between two humans is symmetrized as

$$\mathcal{L}_{\text{pen}}^{\text{total}} = \mathcal{L}_{\text{pen}}(A \to B) + \mathcal{L}_{\text{pen}}(B \to A) \tag{G-6}$$

and can be extended to more than two agents in a straightforward way by summing over all ordered pairs.

Finally, this term is added to the original training objective with a weighting factor $\lambda_{\text{pen}}$(set as $0.1$ in our experiments):

$$\mathcal{L}_{\text{total}} = \mathcal{L} + \lambda_{\text{pen}} \mathcal{L}_{\text{pen}}^{\text{total}} \tag{G-7}$$

Table G-6: Penetration and feet sliding analysis.

| Method | PD(cm)↓ | PFR(%)↓ | FS(%)→ |
|---|---|---|---|
| Ground Truth | $1.740^{\pm.0003}$ | $0.68^{\pm.000}$ | $1.090^{\pm.0006}$ |
| InterMask* | $3.570^{\pm.0002}$ | $10.350^{\pm.0003}$ | $2.630^{\pm.0001}$ |
| DART[†] | $3.240^{\pm.0002}$ | $5.850^{\pm.0003}$ | $2.530^{\pm.0001}$ |
| HINT | $2.652^{\pm.0005}$ | $3.260^{\pm.0009}$ | $1.770^{\pm.0002}$ |
| HINT w $\mathcal{L}_{\text{pen}}^{\text{total}}$ | $\mathbf{2.460}^{\pm.0002}$ | $\mathbf{1.510}^{\pm.0001}$ | $\mathbf{0.910}^{\pm.0001}$ |

Tab. G-6 reports the penetration and feet-sliding metrics of different methods on InterX. PD (Penetration Depth) measures the average depth to which bodies interpenetrate. PFR (Penetration Frame Rate) measures the percentage of frames in which any penetration occurs. FS (Foot Sliding Rate) measures the percentage of frames in which noticeable foot sliding is observed.

The results indicate that HINT already performs competitively without explicit physical constraints, and the physics-enhanced version yields additional improvements.

