# OpenReview forum: "HINT: Hierarchical Interaction Modeling for Autoregressive Multi-Human Motion Generation"
_ICLR.cc/2026/Conference — Submitted to ICLR 2026_

### Official Review · Reviewer_8Sbr · 2025-10-18

**Soundness:** 3
**Presentation:** 3
**Contribution:** 2
**Rating:** 4
**Confidence:** 4

**Summary:**

The paper proposes HINT, an autoregressive framework for generating multi-human motion. The method combines a diffusion model with a sliding-window approach. The authors claim two main contributions: 1) a "Canonicalized Latent Space" that decouples individual motion from inter-person geometry, and 2) a "Hierarchical Motion Condition" strategy that uses local and global conditions to guide the interactivate human motion generation process. The authors claim their method matches offline models and surpasses other autoregressive approaches, achieve improvement on the InterX and InterHuman dataset.

**Strengths:**

1. The paper introduces a Canonicalized Latent Space that encodes each person’s motion in their own local coordinates, rather than in world coordinates.
2. The paper employs a sliding-window, autoregressive generation scheme built on a DiT-based diffusion model.
3. The experiments are comprehensive.

**Weaknesses:**

1. Although the method improves overall interaction motion quality and produces plausible individual motions, it does not explicitly resolve fine-grained physical interactions between bodies in two-person scenarios. As a result, the generated sequences may lack realistic inter-body contact dynamics or exhibit artifacts such as slipping and penetration during close interactions.
2. The multi-human results could be better. The teaser does not clearly demonstrate interactive behaviors among the multiple characters. The motions in the teaser look almost copy–pasted across characters.

**Questions:**

1. The paper claims to support online generation. Has the author considered the transition/smoothing issue when generating different motion sequences continuously?

2. For multi-human motion generation, does inference require providing initial motion (a starting pose/sequence) as input? Will it generate unreasonable human motion if the two people are in the distance initially?

3. The paper states that 3-person and 4-person motions can be generated without additional training. How is this achieved in practice? Can the model ensure that every pair of individuals can interact with each other?

---

> ### Author Response · Authors · 2025-11-23
> **Reviewer 8Sbr (1/3)**
>
> We sincerely thank the reviewer `8Sbr` for the constructive evaluation and positive acknowledgment of our key ideas. We appreciate the recognition of our **Canonicalized Latent Space design**, the **autoregressive sliding-window diffusion framework**, and **the comprehensiveness of our experimental study**. We are grateful for these encouraging comments and address the remaining concerns below.
>
> ---
>
> > #### W1: "Although the method improves overall interaction motion quality and produces plausible individual motions, it does not explicitly resolve fine-grained physical interactions between bodies in two-person scenarios. As a result, the generated sequences may lack realistic inter-body contact dynamics or exhibit artifacts such as slipping and penetration during close interactions."
>
> As explained in "**Rebuttal to All Reviewers**", in HINT, we primarily achieve physical plausibility by explicitly encoding the relative transformations between humans and feeding this information as conditions to the diffusion network, allowing the model to learn the physical dynamics of multi-person interactions **directly from data during training**. We do not explicitly incorporate contact losses or physical collision penalties (e.g., distance-field or SDF-based losses) during training. This is because, with sufficiently large and diverse data, the network can implicitly learn physical regularities—similar to what has been observed in large-scale feature learning (DINOv3 [`R1`]) and video generation systems such as Sora [`R2`].
>
> Nevertheless, our method is fully **compatible with physics-aware losses**. One can simply add an additional physics term to the training objective. We have also included experiments demonstrating that incorporating such losses can further reduce slipping and penetration artifacts. The results (shown below) indicate that HINT already performs competitively without explicit physical constraints, and the physics-enhanced version yields additional improvements.
>
> **Table.** Penetration and feet sliding analysis
>
> | Method                 | PD(cm)↓                      | PFR(%)↓                      | FS(%)→                       |
> | ---------------------- | ---------------------------- | ---------------------------- | ---------------------------- |
> | GT                     | $1.740^{\pm .0003}$          | $0.68^{\pm .000}$            | $1.090^{\pm .0006}$          |
> | InterMask*             | $3.570^{\pm .0002}$          | $10.350^{\pm .0003}$         | $2.630^{\pm .0001}$          |
> | DART†                  | ${3.240}^{\pm .0002}$        | $5.850^{\pm .0003}$          | $2.530^{\pm .0001}$          |
> | HINT                   | ${2.652}^{\pm .0005}$        | $3.260^{\pm .0009}$          | $1.770^{\pm .0002}$          |
> | HINT (+ physical loss) | $\mathbf{2.460}^{\pm .0002}$ | $\mathbf{1.510}^{\pm .0001}$ | $\mathbf{0.910}^{\pm .0001}$ |
>
> **PD** measures the depth of intersection, **PF** measures the percentage of frames with any penetration, and **FS** measures foot–ground sliding.
>
> In summary, although **fine-grained physical interaction modeling is not the primary focus of our work**, the framework can **seamlessly incorporate these physical constraints** when needed. We have made this point more explicit in the revised version.
>
> ---
> **References:**
>
> - [`R1`] Siméoni, Oriane, et al. "Dinov3." *arXiv preprint arXiv:2508.10104* (2025).
> - [`R2`] Liu, Yixin, et al. "Sora: A review on background, technology, limitations, and opportunities of large vision models." *arXiv preprint arXiv:2402.17177* (2024).

---

> ### Author Response · Authors · 2025-11-23
> **Reviewer 8Sbr (2/3)**
>
> > #### W2&Q3: "The multi-human results could be better. The teaser does not clearly demonstrate interactive behaviors among the multiple characters. The motions in the teaser look almost copy–pasted across characters. The paper states that 3-person and 4-person motions can be generated without additional training. How is this achieved in practice? Can the model ensure that every pair of individuals can interact with each other?"
>
> As explained in "**Rebuttal to All Reviewers**", our multi-human results are produced without retraining the model for settings involving more than two agents. As there is currently no public dataset for text-guided motion generation with a variable number of agents, in the main experiments, we directly reuse the two-person HINT model to generate 3- or 4-person motions.
>
> In the two-person formulation, all conditioning signals are defined with respect to the current target agent, and thus remain unchanged when scaling to more agents, except for the partner’s history motion. During training, the partner’s history is padded to a fixed length, and its position is randomly translated to encourage robustness. At test time, for more than two agents, we **concatenate the history motions of all partners** and pad the concatenated vector in the same way.
>
> This simple modification allows the model to operate in multi-human scenarios zero-shot, without architectural changes or additional training. This is the basis of our statement that “3-person and 4-person motions can be generated without additional training.”
>
> However, because the training data contains only two interacting individuals, the model only learns to distinguish between "the target person" and "the other person". When more than two agents are present, this assumption no longer holds, and the model **cannot guarantee correct pairwise role identification**.
>
> As a result, the model sometimes treats all partners as an undifferentiated “aggregate partner”,  which can lead to mirrored or repetitive behaviors, as observed in the teaser. Therefore, while the model can maintain global layout and basic coordination, it cannot strictly ensure that every pair of individuals interacts meaningfully.
>
> To address this limitation, we extend the HOI-M³ dataset to HOI-M$^{3*}$ by **providing atomic, role-specific textual annotations** (e.g., “Person0 sits on sofa and talks to Person1,’’). HOI-M³* includes 52 sequences (~6 minutes each), 1.1M frames, and 1919 sentences, covering 2–5-person interactions. These annotations encode **explicit role identities**, enabling role-aware conditioning for diffusion models. The results finetuned on HOI-M$^{3*}$ shown as below and visualization results (as shown in the supplementary material under `rebuttal_demo/multi_human_generation`) shows that our framework, without architectural change, can support variable numbers of agents and produce more distinct, non-repetitive multi-human interactions.
>
> **Table.** Results on HOI-M$^{3*}$
>
> | Method |  R@Top3$\uparrow$  |  FID$\downarrow$   | MM Dist$\downarrow$ | Diversity$\rightarrow$ |
> | :----: | :----------------: | :----------------: | :-----------------: | ---------------------- |
> |   GT   | $0.881^{\pm .002}$ | $0.001^{\pm .003}$ | $7.176^{\pm .001}$  | $6.809^{\pm .034}$     |
> |  HINT  | $0.460^{\pm .004}$ | $1.501^{\pm .009}$ | $8.202^{\pm .004}$  | $6.849^{\pm .042}$     |

---

> ### Author Response · Authors · 2025-11-23
> **Reviewer 8Sbr (3/3)**
>
> > #### Q1: "The paper claims to support online generation. Has the author considered the transition/smoothing issue when generating different motion sequences continuously?"
>
> In our framework, continuity is handled by **overlapping sliding** windows, where each new window reuses a segment of the previously generated frames as history, and all relative transforms (root orientation/translation) are computed from these actually generated frames, rather than being re-predicted, which prevents error amplification across windows. Although InterHuman and InterX mainly contain relatively short sequences, we further tested our pipeline on a longer HOI-M³ subset HOI-M³\*, where continuous rollouts show no noticeable jumps or discontinuities at window boundaries and motions evolve smoothly over time. Furthermore, long-horizon visualizations also displays the smooth transition. (as shown in the supplementary material under `rebuttal_demo/long_seq`)
>
> As summarized in the table below, we quantitatively evaluate transition quality on HOI-M³\*. Since HOI-M³\* contains a variable number of humans, extending InterMask* and other offline baselines to this setting is non-trivial. Therefore, we report comparisons only against DART$^{\dagger}$ on this dataset.
>
> **Table.** Transition evaluation on HOI-M$^{3*}$
>
> |      Method      |       FID$\downarrow$       |   Diversity$\rightarrow$    |       PJ$\rightarrow$       |       AUJ$\downarrow$       |
> | :--------------: | :-------------------------: | :-------------------------: | :-------------------------: | :-------------------------: |
> |        GT        |     $0.001^{\pm .000}$      |     $6.135^{\pm .040}$      |     $0.169^{\pm .001}$      |     $0.001^{\pm .001}$      |
> | DART$^{\dagger}$ |     $12.032^{\pm .026}$     |     $5.428^{\pm .048}$      | $\mathbf{0.413}^{\pm .003}$ | $\mathbf{0.252}^{\pm .001}$ |
> |       HINT       | $\mathbf{1.305}^{\pm .021}$ | $\mathbf{6.188}^{\pm .026}$ |     $0.442^{\pm .005}$      |     $0.272^{\pm .005}$      |
>
> FID measures the distribution gap between generated and real motions, where lower values indicate more realistic sequences. Diversity captures how varied the generated motions are, with higher values indicating richer, less mode-collapsed behavior. Peak Jerk (PJ) measures the maximum jerk (i.e., sudden change in acceleration) over all joints during the transition, while Area Under the Jerk curve (AUJ) integrates jerk over time to reflect overall smoothness of the transition. On HOI-M³\*, HINT achieves a much lower FID than DART$^{\dagger}$ and Diversity close to GT, indicating both high realism and variety. Meanwhile, the PJ and AUJ scores of HINT are comparable to those of DART$^{\dagger}$, suggesting that our method maintains  transition smoothness while improving realism and diversity.
>
>
>
> > #### Q2: "For multi-human motion generation, does inference require providing initial motion (a starting pose/sequence) as input? Will it generate unreasonable human motion if the two people are in the distance initially?"
>
>  Our system supports two types of initialization:
>
>   1. **User-provided motion history.** In scenarios that require precise control or integration with external systems, the user can provide a short initial sequence for each human. The model then performs online generation conditioned on this history.
>   2. **Initialization from scratch under text-only conditioning.** When only a textual description is given and no external history is available, we adopt a simple, unified initial pose such as a standing pose or T-pose. The diffusion model then rolls out the full motion sequence from this starting state, guided by the text and interaction design.
>
> And in our current framework, we introduce several interaction-related losses during training (such as Relative Orientation and Distance Map losses) to constrain the evolution of pairwise facing directions and distances, encouraging the model to respect spatial priors observed in real data.
>
> Empirically, we find that when the initial distance between two agents is large, the model typically first generates an approach motion until a reasonable interaction distance is reached, and only then produces the handshake or dance motion, rather than executing the instruction at an implausible spatial separation. (as shown in the supplementary material under `rebuttal_demo/far_initial_distance`)
> We have clarified the initialization settings in the revised version.
>
> ---
> We would like to sincerely thank Reviewer `8Sbr` again for the valuable time and constructive feedback provided during this review.

---

> > ### Comment · Reviewer_8Sbr · 2025-11-24
> >
> > Thank you for your comprehensive responses. They have sufficiently addressed my concerns and clarified the issues I raised.
> > After considering the other reviewers’ comments and your rebuttal, I have decided to increase my score.

---

> > > ### Author Response · Authors · 2025-11-24
> > >
> > > We sincerely thank the reviewer for the thoughtful assessment and for increasing the score after considering our clarifications. We truly appreciate your time and constructive feedback, which have helped strengthen the paper.

---

### Official Review · Reviewer_wSXb · 2025-10-19

**Soundness:** 2
**Presentation:** 3
**Contribution:** 2
**Rating:** 4
**Confidence:** 4

**Summary:**

The paper tackles multi-human motion generation, an important yet comparatively underexplored area in generative AI. Compared to single-human generation, multi-human settings add significant complexity due to coordination, spatial relations, and temporal consistency across agents. The authors’ decoupling strategy aims to enable spatially aware, time-consistent generation across multiple people. While diffusion models are often constrained to fixed lengths, token-based methods (e.g., VQ-VAE) enable variable-length synthesis; this breaches traditional diffusion-based methods by adopting a sliding-window procedure for continuous generation. Results on established benchmarks appear promising. However, the qualitative evidence shows limited improvements in genuine interactions: most examples are generic, non-contact scenarios with relatively simple motions and limited inter-agent coupling.

**Strengths:**

* Clear motivation for addressing multi-human motion, a topic that demands richer spatial/temporal modeling than single-human generation.
* A principled decoupling/conditioning design intended to preserve local motion quality while handling inter-person relations.
* Practical use of a sliding window to extend diffusion beyond fixed-length clips, enabling continuous rollouts.
* Competitive results on standard benchmarks.

**Weaknesses:**

* **Unspecified text-to-length mapping.** Global conditioning uses the “total frame number” ($T_N$) *according to the textual description*, but the paper does not explain how ($T_N$) is inferred or parsed from text at test time. This is central to the **compositional command** narrative and should be made explicit.

* **Canonicalization and drift across windows.** The approach removes absolute placement and re-injects pairwise transforms. The paper should discuss **how relative transforms are obtained and propagated** (predicted vs. read from history) and whether compounding errors arise across windows—especially for long sequences and when ($N$>2).

* **Limited contribution of word-level text.** The ablation suggests removing word-level embeddings does not change R@Top3 (0.672 $\rightarrow$ 0.672), which weakens the claim that token/word-level guidance is essential for semantic fidelity. This discrepancy should be analyzed and reconciled.

* **Underpowered user study.** With only ~15 participants and minimal protocol detail, the subjective evaluation provides limited evidentiary value for the broader claims.

* **Interaction richness is not demonstrated.** While the system can produce multiple agents moving concurrently, the evidence for **true, contact-rich interactions** (touch, handoffs, coordinated manipulation) is thin. The demos often look like parallelized single-person motions rather than tightly coupled multi-person behaviors.

**Questions:**

1. **Number of agents from text.** Can the model **infer** how many humans to generate purely from the text description? If yes, how is this determined? If not, what is the intended interface for specifying (N)?

2. **Initialization / first frames.** Does generation require an initial history (e.g., the first frame or a short prefix) for each agent? If text-only generation is supported, please describe the initialization procedure.

3. **Role control for (N $\geq$ 3).** How does the system distinguish and control different people—especially three or more—when a single prompt describes different roles? Is the model capable of **per-agent conditioning** simply based on text prompts?

4. **Inter-penetration and physical plausibility.** Does the method incorporate **hard constraints** (e.g., collision checks) or rely solely on training penalties? Can the authors provide **contact-rich** examples (e.g., one person touches another’s shoulder or knee; passing an object) and report basic contact/collision metrics?

5. **Spatial-text reasoning.** If the text says “two people shake hands” but the initial distance is large, will the agents **walk toward each other before shaking hands**, or do they attempt to satisfy the prompt without reconciling the spatial precondition?

6. **Out-of-distribution prompts.** How does the model behave on **outside-the-scope** or complex motions not well represented in the training set? The provided demos appear elementary; please comment on robustness and failure modes.

---

> ### Author Response · Authors · 2025-11-23
> **Reviewer wSXb (1/4)**
>
> We sincerely thank the reviewer `wSXb` for the thoughtful evaluation, constructive feedback. We appreciate the reviewer’s recognition of **the importance of multi-human motion generation**, the motivation behind our **decoupling strategy**, the **principled conditioning design**, and the **practicality of our sliding-window diffusion framework**. Below we address all concerns in detail.
>
> ---
>
> > #### W1:" Unspecified text-to-length mapping."
>
> $T_N$ is directly provided by the **dataset**. Similar to **offline** generation methods, which generate the entire sequence using the ground-truth length, we also use the ground-truth sequence length as the total generation length for a fair comparison.
> However, in real applications, $T_N$ can be determined in several practical ways,
>
> 1. **User-controlled** based on instruction complexity. Empirically, users can assign a suitable $T_N$ according to the complexity of the textual command.
> 2. Using **a large $T_N$ and trimming afterward**. One may set $T_N$ to a sufficiently large value and then manually trim the generated video once the intended action has been completed.
> 3. **Automatic stopping** with a language–motion similarity metric (Our Eval Model). A more advanced option is to set $T_N$ to a relatively large upper bound and use the evaluation model we employ for computing MM-Dist. This model measures the similarity between the generated motion and the given textual description, and generation can stop as soon as the similarity surpasses a predefined threshold.
>
> > #### W2: "Canonicalization and drift across windows."
>
> We clarify our design for the canonicalization and drift across windows as follows:
>
> First, the relative transforms are **computed from history**, not re-predicted. In our framework, the relative rotations and translations used in each window are computed directly from the actual generated frames of the previous window (based on SMPL root pose and joint positions), rather than being predicted again by the network. Thus, the relative geometry is not recursively updated by noisy predictions and does not amplify model error across windows.
>
> Second, the drift across windows are controlled as follows,
>
>   1. Short-window latent prediction reduces error propagation. Rather than rolling out frame by frame, HINT operates in a canonicalized latent space and predicts future motion in short windows (history length $H$, future length $K$). Each step only propagates errors at the **window level**, and the diffusion process refines a coherent latent trajectory within each window, which empirically stabilizes long-horizon rollouts.
>   2. Hierarchical conditions and canonicalized latent space prevent drift. Local conditions enforce **short-term dependencies and fine-grained semantic alignment ** inside each window, while global conditions anchor each window to the overall script, **preventing long-term semantic drift**. In addition, encoding motion in canonicalized per-person coordinates while feeding global geometry as explicit relative transforms decouples global position from motion semantics, reducing the amplification of small pose errors over time.
>  3. Drift is evaluated via long-horizon **qualitative analysis**. In our online setting, there is no fixed absolute ground-truth trajectory for arbitrarily long rollouts, so cumulative error is difficult to quantify by GT distance alone. Instead, we inspect long-horizon visualizations to check for systematic drift, In our multi-person long-sequence experiments, we do not observe such trends, the interactions remain stable over time.(as shown in the supplementary material under `rebuttal_demo/long_seq`)
>  4. Autoregressive generation allows **interactive correction**. Since the system is autoregressive and user-controllable, in real applications, small deviations can be corrected in subsequent windows by issuing lightweight steering commands (e.g., adjusting position or direction). This interactive loop further mitigates the practical impact of cross-window error accumulation.

---

> ### Author Response · Authors · 2025-11-23
> **Reviewer wSXb (2/4)**
>
> > #### W3: "Limited contribution of word-level text."
>
> It is correct that removing the word-level embedding leads to almost no change in R@Top3, but this does not imply that token-level guidance is unimportant. Rather, it reflects the characteristics of the metric itself.
> When the word-level embedding is removed, the global semantic alignment of a motion clip is still largely maintained by the compositional command embedding, which stabilizes text–motion retrieval scores such as R@Top3. However, the purpose of the word-level embedding is to guide **fine-grained, within-window semantics**, subtle action details and temporal phrasing that do not strongly affect retrieval metrics but do influence the realism and distributional fidelity of the generated motion.
>
> Consistently, we observe a clear degradation in realism metrics: removing the word-level embedding **increases FID from 3.100 to 3.295**, indicating a shift in the generated distribution and a loss of local semantic precision. Thus, the unchanged R@Top3 and the contribution of word-level guidance are not contradictory, Word-level embedding improves fine-grained semantic faithfulness and visual quality, reflected by FID degradation.
>
> We have clarified this distinction in the revised version to avoid conflating global semantic alignment with local semantic fidelity.
>
> > #### W4: "Underpowered user study."
>
> We thank the reviewer for the careful assessment of our user study. We agree that a sample size of 15 participants may limit the statistical strength of subjective evaluation. At the time of submission, we were only able to collect 15 valid user-study responses. We have since gathered additional responses, bringing the total to 50. Additionally, our key conclusions in the paper are supported by objective quantitative metrics (e.g., FID, R@Top3) and systematic ablations, while the user study is intended only to verify whether human perception is consistent with these quantitative trends.
>
> > #### Q1: "Number of agents from text. "
>
> In our **quantitative experiments** in the paper, the number of agents is **fixed to two** because existing text-guided multi-human motion-generation datasets only support two-person interactions. In such cases, the model can already infer the number of agents directly from the textual description. For example, a prompt such as “one person walks to the other person, and they shake hands’’ implicitly specifies two people.
>
> When extending from 2 agents to 2–5 agents, i.e. in our newly annotated HOI-M$^{3*}$ dataset, we adopt "structured textual descriptions". The number of sentences naturally determines the number of target individuals to be generated. Each sentence corresponds to one generated motion sequence. For instance, in the description “Person0 sits on the bed, then walks to the sofa, and puts their hands on Person1’s shoulders,’’ the first token, Person0, indicates the target person. When generating Person1’s motion, the corresponding sentence is “Person1 sits on the sofa, and Person0 puts hands on Person1’s shoulders.’’Since HOI-M$^{3*}$ provides atomic textual descriptions for each individual, the model does not need to explicitly reason about how many agents are involved. Below we shows the results on HOI-M$^{3*}$.
>
> **Table.** Results on HOI-M$^{3*}$
>
> | Method |  R@Top3$\uparrow$  |  FID$\downarrow$   | MM Dist$\downarrow$ | Diversity$\rightarrow$ |
> | :----: | :----------------: | :----------------: | :-----------------: | ---------------------- |
> |   GT   | $0.881^{\pm .002}$ | $0.001^{\pm .003}$ | $7.176^{\pm .001}$  | $6.809^{\pm .034}$     |
> |  HINT  | $0.460^{\pm .004}$ | $1.501^{\pm .009}$ | $8.202^{\pm .004}$  | $6.849^{\pm .042}$     |
>
> Additionally, if the input textual prompt is not already in a structured format, a language model such as **ChatGPT** can be used to **convert it into the required structured form prior** to generation.
>
> > #### Q2: "Initialization / first frames. Does generation require an initial history (e.g., the first frame or a short prefix) for each agent? If text-only generation is supported, please describe the initialization procedure."
>
> Our system supports two types of initialization:
>
> 1. **User-provided motion history.** In scenarios that require precise control or integration with external systems, the user can provide a short initial sequence for each human. The model then performs online generation conditioned on this history.
> 2. **Initialization from scratch under text-only conditioning.** When only a textual description is given and no external history is available, we adopt a simple, unified initial pose such as a standing pose or T-pose. The diffusion model then rolls out the full motion sequence from this starting state, guided by the text and interaction design.
>
> We have clarified these two initialization modes in the revised version.

---

> ### Author Response · Authors · 2025-11-23
> **Reviewer wSXb (3/4)**
>
> > #### Q3: "Role control for ($N \geq 3$).How does the system distinguish and control different people—especially three or more—when a single prompt describes different roles? Is the model capable of per-agent conditioning simply based on text prompts?"
>
> As explained in "**Rebuttal to All Reviewers**", In HINT, we demonstrate that a motion-generation network trained solely on two-person datasets can be directly extended to multi-person scenarios without any additional retraining.
>
> This approach, however, introduces challenges in **role identification**. Since the model is trained on datasets containing only two interacting individuals, it only needs to distinguish between the “target person’’ and “the other person.’’ This assumption breaks down when the number of agents exceeds two; the model can no longer ensure proper pairwise interactions among all individuals, leading to some repetitive behaviors, as shown in Fig1 and supplementary videos.
>
> To address this issue, we extend HOI-M3 to HOI-M3* by annotating structured textual descriptions for each human. The number of sentences naturally corresponds to the number of target individuals to be generated. Therefore, the role identification is directly handled via **standardizing the input text**.
>
> > #### W5&Q4: "Inter-penetration and physical plausibility. Does the method incorporate hard constraints (e.g., collision checks) or rely solely on training penalties? Can the authors provide contact-rich examples (e.g., one person touches another’s shoulder or knee; passing an object) and report basic contact/collision metrics?"
>
> In **Fig. C-4, C-5, C-7, and C-9**, as well as in the additional examples provided in the rebuttal(as shown in the supplementary material under `rebuttal_demo/interaction`), we show several close-range interaction cases, demonstrating that our model can learn multi-human interaction patterns directly from data. As noted in "**Rebuttal to All Reviewers**", some penetration may still arise because our method does not impose explicit physical constraints.
>
> **1. Use of physical constraints.**
>  HINT does **not** incorporate hard physical constraints such as collision checks, SDF-based penalties, or explicit contact supervision during training. Instead, we rely on **data-driven learning of interaction regularities**, guided by explicit encoding of **pairwise relative transformations** between humans. This provides strong geometric cues to the diffusion network and enables the model to implicitly learn stable human-human dynamics. Similar to findings in large-scale representation learning (e.g., DINOv3 [`R1`]) and video generation (e.g., Sora [`R2`]), sufficiently diverse data can encourage the model to internalize basic physical regularities even without hand-crafted physical losses.
>
> **2. Compatibility with physics-aware losses.**
> Importantly, our framework is **fully compatible** with additional physics-aware terms. One may incorporate standard collision or contact losses by simply adding them to the objective. For completeness, we also trained a variant with such losses and observed that it further suppresses penetration and slipping artifacts.
>
> **3. Quantitative evaluation.**
> We provide a detailed analysis using **penetration depth**, **penetration frame rate**, and **foot sliding rate**. The results (shown below) indicate that HINT already performs competitively without explicit physical constraints, and the physics-enhanced version yields additional improvements.
>
> **Table.** Penetration and feet sliding analysis
>
> | Method                 | PD(cm)↓                      | PFR(%)↓                      | FS(%)→                       |
> | ---------------------- | ---------------------------- | ---------------------------- | ---------------------------- |
> | GT                     | $1.740^{\pm .0003}$          | $0.68^{\pm .000}$            | $1.090^{\pm .0006}$          |
> | InterMask*             | $3.570^{\pm .0002}$          | $10.350^{\pm .0003}$         | $2.630^{\pm .0001}$          |
> | DART†                  | ${3.240}^{\pm .0002}$        | $5.850^{\pm .0003}$          | $2.530^{\pm .0001}$          |
> | HINT                   | ${2.652}^{\pm .0005}$        | $3.260^{\pm .0009}$          | $1.770^{\pm .0002}$          |
> | HINT (+ physical loss) | $\mathbf{2.460}^{\pm .0002}$ | $\mathbf{1.510}^{\pm .0001}$ | $\mathbf{0.910}^{\pm .0001}$ |
>
> **PD** measures the depth of intersection, **PF** measures the percentage of frames with any penetration, and **FS** measures foot–ground sliding.
>
> **In summary**, while explicit physical modeling is not the central focus of HINT, our design naturally supports it and can be enhanced with physics-based losses when needed.
>
> ---
> **References:**
>
> - [`R1`] Siméoni, Oriane, et al. "Dinov3." *arXiv preprint arXiv:2508.10104* (2025).
> - [`R2`] Liu, Yixin, et al. "Sora: A review on background, technology, limitations, and opportunities of large vision models." *arXiv preprint arXiv:2402.17177* (2024).

---

> ### Author Response · Authors · 2025-11-23
> **Reviewer wSXb (4/4)**
>
> > #### Q5: "Spatial-text reasoning. If the text says “two people shake hands” but the initial distance is large, will the agents walk toward each other before shaking hands, or do they attempt to satisfy the prompt without reconciling the spatial precondition?"
>
> In our current framework, we introduce several interaction-related losses during training (such as Relative Orientation and Distance Map losses) to constrain the evolution of pairwise facing directions and distances, encouraging the model to respect spatial priors observed in real data.
>
> Empirically, we find that when the initial distance between two agents is large, the model typically first generates an approach motion until a reasonable interaction distance is reached, and only then produces the handshake motion, rather than executing the instruction at an implausible spatial separation. (as shown in the supplementary material under `rebuttal_demo/far_initial_distance`)
>
> > #### Q6: "Out-of-distribution prompts. How does the model behave on outside-the-scope or complex motions not well represented in the training set? The provided demos appear elementary; please comment on robustness and failure modes."
>
> HINT is capable, to some extent, of handling OOD text and generating plausible motions. There are two main reasons for this:
>
> 1. **Language generalization via CLIP**: CLIP maps semantically similar actions (shake hands, clap hands) to nearby vectors, allowing the model to interpret previously unseen textual descriptions in a meaningful way.
> 2. **Learned interaction priors**: Our network has learned certain physical patterns of human-human interaction, which can be applied to novel actions beyond the training set.
>    We have provided an illustrative example demonstrating this capability.(as shown in the supplementary material under `rebuttal_demo/long_seq`) In this video, the texts are generated by the LLM to demonstrate the capability to handle OOD text.
>
> Of course, this ability **remains limited** at present due to **the scarcity of training data**. Achieving truly open-vocabulary motion generation will require larger and more diverse datasets. To this end, we are currently annotating HOI-M³* to expand the available data.
>
> ---
>
> We would like to sincerely thank Reviewer `wSXb` again for the valuable time and constructive feedback provided during this review.

---

> > ### Comment · Reviewer_wSXb · 2025-11-25
> >
> > I thank the authors for their detailed rebuttal and the extensive additional experiments and visualizations. I particularly like the metrics quantifying interpenetration and the utilization of the $\text{HOI-M}^{3\ast}$ dataset. I believe these should also be considered as core evaluations in future works regarding multi-human motion generation.
> >
> > That said, my overall assessment remains largely unchanged. While the new experiments strengthen the empirical section, I still see the contribution as fairly incremental relative to prior multi-human and autoregressive motion works. The hierarchical interaction module (VQVAE+relative feature), canonicalized latent space, and sliding-window conditioning strike me as a reasonable combination of existing ideas rather than a qualitatively new way of modeling multi-human interactions. The gains, although measurable, do not convincingly demonstrate a step change in how we approach text-driven multi-human motion generation.
> >
> > Moreover, even with the additional demos provided in the rebuttal, the qualitative results remain unconvincing to me. The generated interactions often look limited in richness and realism, especially when the number of agents grows (though I totally understand that multi-human is just an extension of the current framework). This gap between the strong claims in the paper and the actual visual quality makes me hesitant to view the method as a practically impactful solution to this problem.
> >
> > In summary, I appreciate the authors’ efforts and the improved analysis, but I would remain my score.

---

> > > ### Author Response · Authors · 2025-11-26
> > >
> > > Thank you. We are happy that you acknowledge the value of our newly added penetration metrics, visualizations, and extended experiments.
> > >
> > > First, we would like to clarify several key aspects of our contributions. Our framework does not rely on VQ-VAE+relative feature design mentioned by the reviewer. Instead, we adopt VAE + diffusion architecture that adheres to a sliding-window paradigm. While this pipeline is employed in single-human autoregressive generation(DART [`R1`]), our design is explicitly task-oriented toward handling a variable number of humans. To this end, we introduce a canonicalized latent space, where each human is represented in their own local coordinate frame while inter-person geometry is captured through relative transformations. This decouples individual motion semantics from group configuration and allows the same representation to scale seamlessly as agents are added or removed. Moreover, we systematically analyze interaction-related conditions and integrate them into a hierarchical conditioning scheme. At the local level (within each sliding window), conditions on the target human’s history, partner history, step index, and word-level text capture fine-grained temporal and social dependencies. At the global level (across sliding windows), conditions on the sequence index, total frame number, and compositional command embedding maintain long-term temporal consistency. Ablation studies show that both levels are essential, removing either group significantly degrades the stability of multi-human long-horizon generation. To our knowledge, this is the first framework capable of generating text-driven motions for a varying number of humans within one unified model.
> > >
> > > Second, we would like to point out that the community currently lacks datasets that provide text-to-motion supervision for variable-number, richly interactive human groups. Existing public datasets mainly cover everyday activities with moderate interpersonal spacing, and close-contact interactions are relatively rare. We carefully searched for suitable multi-human resources and further extended and re-annotated the HOI-M$^{3}$ dataset to better approximate variable-human interaction scenarios. Although even with these efforts, the overall interaction richness is still limited by the available data. Nevertheless, under these realistic constraints, our method consistently outperforms available online baselines (Tab. 1 and Tab. A-1), demonstrating clear advantages.
> > >
> > > Finally, we thank the reviewer again for the careful evaluation. For future research directions, we think that constructing more specifically designed datasets may be a potential solution to the challenges in multi-human generation.
> > >
> > > ---
> > >
> > > **References:**
> > >
> > > [`R1`] Kaifeng Zhao, et al. Dartcontrol: A diffusion-based autoregressive motion model
> > > for real-time text-driven motion control. *International Conference on Learning Representations*
> > > *(ICLR)*, 2024.

---

### Official Review · Reviewer_ArWa · 2025-10-27

**Soundness:** 3
**Presentation:** 3
**Contribution:** 3
**Rating:** 6
**Confidence:** 3

**Summary:**

This paper proposes a novel hierarchical interaction modeling approach that effectively addresses the entanglement of global positions and individual motion semantics in multi-person motion generation. By decoupling each person's motion into a local space and using multi-person interaction information as a condition for diffusion, the method ensures semantic clarity. Additionally, an online sliding-window strategy built upon this framework guarantees efficient online generation. Experiments validate the effectiveness of both HMC and online generation.

**Strengths:**

1. This paper insightfully identifies a potential obstacle to generalizing two-person interaction motion generation to larger groups—namely, the entanglement of global information—and proposes and validates a decoupling strategy, thereby demonstrating strong originality and clearly establishing the significance of the problem.

2. The exposition is clear; the authors’ ideas are easy to grasp.

3. The supplementary videos present qualitative multi-person generation results and experiments that confirm the successful extension of HMC to groups larger than two.

**Weaknesses:**

1. Extracting interaction information into the global diffusion pipeline may lengthen the conditioning vector. Extending a two-person scenario to three is still manageable, but scaling to larger groups becomes problematic. Taking agent A’s viewpoint as an example, the paper encodes partner history by using B’s rotation $R$ and translation $T$ relative to A. Section 3.5 further suggests that expanding from two to N people simply means concatenating the partner-history embeddings of all additional partners; therefore, the history length—and hence the condition size—grows exponentially with the number of agents.

2. Multi-person interaction still relies on fine-grained user control. As noted in line 286 (Section 3.3, Global Conditions, Compositional Command Embedding), if the user supplies a full description that contains multiple step-by-step instructions, a sentence-level global token is provided to guide generation. **It remains unclear whether performance drops noticeably when such a global prompt is not given.**

**Questions:**

1. **Has HINT ever attempted to generate longer clips, such as results lasting one minute** (possibly stitched from shorter segments), with corresponding textual instructions possibly produced by an LLM? Generating long-duration motions can test motion coherence and thereby demonstrate how the sliding-window strategy copes with the challenges of long-term sequence generation.

2. Yet if the user’s instruction is a vague one like “A person first shakes hands with the person opposite them and then hugs the others,” it should still be possible to infer who is “opposite” and who are “the others” by using global information about all agents, and then to generate the correct motion given the prior history. **By converting global information into relative information for each individual agent, does HMC disrupt the understanding of the implicit global semantics?**

---

> ### Author Response · Authors · 2025-11-23
> **Reviewer Arwa (1/2)**
>
> We sincerely thank the reviewer `ArWa` for the positive and encouraging assessment, and for recognizing the value of **disentangling global positions from individual motion semantics**, the originality of our **hierarchical interaction modeling**, **the clarity of our exposition**, and the **scalability** to multi-person settings shown in the supplementary results. We address the remaining concerns below.
>
> ---
>
> > #### W1: “Scaling partner-history encoding to more agents may cause the conditioning vector to grow rapidly, since histories from all partners are concatenated.”
>
> We would like to clarify that, **for each human**, the conditioning vector in our framework grows **linearly**  with respect to the number of agents.
>
> For any target agent A, we only include the relative historical geometry of each of the other $N-1$ partners (i.e., their relative rotations and translations expressed in A’s coordinate frame). Thus, the conditioning size for a single agent increases as **$O(N)$**. When extending from 2 to 3, 4, or 5 humans, the diffusion backbone remains unchanged; only the valid input length of the conditioning branch for each agent grows linearly with the group size.
>
> In contrast, many offline generation methods concatenate all agents’ inputs or latent codes, causing both **the condition vector and the model architecture to grow rapidly as the number of humans increases**, which leads to significant increases in parameter count and GPU memory usage. From this perspective, our design is substantially **more lightweight and scalable**.
>
> Additionally, if one wishes to further limit the conditioning size in large-crowd scenarios, our framework naturally supports simple heuristics:
>
>  - selecting only the **K nearest partners** (local interactions are usually the most relevant);
>  - or using an **LLM** to **infer which subset of people are semantically involved in the input text**, and constructing conditions only for those relevant partners.
>
> Thus, the condition size depends on the actual interaction set rather than the total number of humans. This aligns more closely with real-world multi-human interaction patterns.
>
> In summary, our method keeps the diffusion backbone independent of group size, grows only linearly on the conditioning side, and allows further partner selection when needed. Therefore, it does not suffer from exponential growth of the conditioning vector.
>
> > #### W2: "Multi-person interaction still relies on fine-grained user control."
>
> In multi-human interaction modeling, fine-grained user control is **indeed necessary** for our setting. As an autoregressive method, HINT requires step-by-step textual instructions, such as “one person puts both hands on the other person’s shoulders, then claps hands, and jumps’’, to guide motion generation over time. Even if this global prompt is **not provided**, the model can still rely on the **history motions** of all agents together with the **word-level text tokens** in the local conditions to produce plausible motions, although with a slight degradation in performance.
>
> Our ablation study in Tab. 2 shows that removing this Compositional Command Embedding leads to an increase in FID from 3.100 to 3.341, while R@Top3 degrades from 0.672 to 0.669). This suggests that, although token-level text alone can preserve basic semantic consistency, the absence of a sentence-level semantic anchor can **weaken the stability of the generated distribution and the coherence across segments**, which is reflected in long-horizon consistency metrics. Thus, we view the sentence-level global embedding primarily as a mechanism that helps **organize action phases and improves cross-window stability**.
>
> In fact, according to ablation study in Table 2, providing either the word-level text embedding or the compositional command embedding alone is sufficient for HINT to generate reasonable motions. We also provide a supplementary video of one-minute long to show that HINT can handle long compositional commands.(as shown in the supplementary material under `rebuttal_demo/long_seq`)

---

> ### Author Response · Authors · 2025-11-23
> **Reviewer Arwa (2/2)**
>
> > #### Q1: "Longer clips generation with textual instructions produced by an LLM."
>
> We conducted an additional experiment to evaluate the temporal consistency of HINT over a longer time horizon. Specifically, without modifying the architecture, we used the sliding-window strategy to continuously generate motion for about one minute (>1800 frames). To avoid human bias in designing the scenario, we use **multi-step, structured long textual instructions generated by LLM** ( "The two people stand facing each other and wave their hands in greeting; The two people stand there while nodding; The two people stand and chat with relaxed hand gestures and head movements; One person points to the side while the other turns and looks in that direction; The two people laugh together, slightly leaning their upper bodies forward and back; One person demonstrates a small arm movement and the other imitates it in sync; One person practice a simple side-step, while the other stands; One person gently pats the other on the shoulder as they continue talking; The two people swap positions by slowly circling around each other; The two people face each other again and repeat the synchronized side-steps; The two people bow politely to each other at the waist and return upright;  The two people step back together while still facing each other and waving goodbye; Finally, the two people turn slightly away from each other and walk off in opposite directions.") to drive the entire sequence.
> As shown in the video, the model is able to stably roll out long-horizon motions without noticeable drift, mode collapse, or structural misalignment between characters, indicating that the sliding-window mechanism maintains good temporal consistency over extended durations.(shown in the supplementary material under `rebuttal_demo/long_seq`)
>
> > #### Q2: "By converting global information into relative information for each individual agent, does HMC disrupt the understanding of the implicit global semantics?"
>
> Converting global information into each agent’s local coordinate frame in HMC does not remove the information needed to express such semantics. We use rigid affine transformation to transform all conditions into the local coordinate of the target human. Absolute world coordinates are discarded, but **relative orientation, and distance between humans** are fully preserved, which are precisely the cues used to infer who is “in front” or “opposite”. This formulation also aligns with how humans typically interpret spatial relations, we reason about “the person opposite me” based on relative geometry rather than an absolute global origin. As a result, the canonicalization step makes motion representations cleaner and more shared across humans without disrupting implicit global semantics.
>
> For example, in our qualitative results (as shown in the supplementary material under `rebuttal_demo/relative_trans`), the red human first correctly identifies the person directly opposite him and waves to that partner, and then turns to the other one, indicating that converting global information into relative information does not disrupt the understanding of implicit semantics.
>
> ---
>
> We would like to sincerely thank Reviewer `ArWa` again for the valuable time and constructive feedback provided during this review.

---

> > ### Comment · Reviewer_ArWa · 2025-11-24
> > **Good Luck**
> >
> > As I mentioned before, this is a sufficiently elegant and impressive piece of work. The author's explanation that multi-person information does not increase computational complexity has dispelled my concerns. While the long video generation results show relatively obvious traj drift in the later segments, the global information of the actions generated in the early stage remains relatively correct, and the instruction-following performance is quite good. Based on the author's response, I have decided to increase my confidence to 4. Good luck!

---

> > > ### Author Response · Authors · 2025-11-24
> > >
> > > We are grateful for your thoughtful evaluation and supportive feedback. Thank you!

---

### Official Review · Reviewer_uvFZ · 2025-10-30

**Soundness:** 3
**Presentation:** 2
**Contribution:** 2
**Rating:** 4
**Confidence:** 4

**Summary:**

HINT is an autoregressive, diffusion-based framework for online multi-human motion generation that maps each person’s motion into a canonicalized latent space and synthesizes future frames with a sliding-window process guided by hierarchical local/global conditions, enabling variable-length sequences and easy scaling to more agents.  It reports strong realism, achieving FID 3.100 on InterHuman and 0.278 on InterX, outperforming online baselines while remaining slightly behind the best offline method in text alignment.  Ablations confirm the necessity of the canonicalized latent space and both condition tiers.  Inference per window is about 1.1 s on a single 3090 GPU (similar to InterMask* and slower than DART†), with overall results validating the approach’s effectiveness for streaming, multi-agent motion.

**Strengths:**

- Clear decomposition (canonicalization + hierarchical conditioning) that makes variable-length, multi-agent generation straightforward to implement.
- Solid quantitative improvements on realism (FID) with extensive ablations isolating the contribution of each condition.
- Simple path to >2 agents without retraining (shared weights; partner-history concatenation).

**Weaknesses:**

- The paper’s problem setting seems already addressed by prior MDM extensions (e.g., priorMDM);
- Novelty feels incremental—canonicalization + sliding window + standard diffusion conditioning.

**Questions:**

- What is new beyond prior MDM-based streaming/online variants (e.g., priorMDM)? Please add a direct, apples-to-apples comparison and clarify conceptual differences.
- How do you control error accumulation across windows?

---

> ### Author Response · Authors · 2025-11-23
> **Reviewer uvFZ (1/2)**
>
> We sincerely thank Reviewer `uvFZ` for the thoughtful evaluation, constructive questions, and positive recognition of our framework’s **decomposition, empirical results, and multi-human extensibility**. Below we address all concerns in detail.
>
> ----
>
> > #### Q1. “HINT vs. priorMDM”
>
> PriorMDM[`R1`] is an offline method designed for single-human motion generation. It also introduces two potential extensions, DoubleTake and ComMDM, which attempt to extend priorMDM toward long-horizon and two-human generation. However, these two extensions still operate under **fixed-number, offline generation** settings, and are not designed for **varying numbers of humans nor online long-horizon synthesis**. In particular:
>
> **• DoubleTake (long-horizon extension)**：the long sequence is split into short chunks, generated independently, and refined only at transition boundaries. As a result, DoubleTake does not support streaming, and online generation.
>
> **• ComMDM (two-human extension)** requires a **separately trained communication block** tied to exactly two humans, when the number of agents changes, the block must be redesigned and retrained. So ComMDM does not generalize to variable agent counts or long-horizon online scenarios.
>
> Our work is not a direct adaptation of single-person autoregressive models. Instead, we follow a task-oriented motivation to design a method that can naturally handle a varying number of agents and long-horizon generation. Although canonicalization + sliding window + conditioning is typical in single-human autoregressive generation, HINT introduces non-trivial improvements at every stage of this pipeline, enabling the overall process to scale effectively to multi-agent, long-sequence scenarios.
>
> ##### **Key conceptual contributions beyond priorMDM**
>
> **1. Ego-centric canonicalization (not global-frame canonicalization)**
> Prior single-/two-human generation methods normalize motion in a shared global frame. In multi-human (>2) settings, this tightly couples absolute coordinates with relative geometry, leading to drift and unstable interactions.
>
> We introduce per-agent ego-centric canonicalization, where each human is encoded/generated in its own **local coordinate frame** and interactions are captured via explicit relative transformations. This design is inherently **independent of agent count** and forms a **principled, scalable foundation**.
>
> **2. Online sliding-window generation (vs. two-stage refinement)**
> Unlike DoubleTake’s “segment + refinement” pipeline, we perform **single-stage, streaming generation**, where each window directly conditions on the generated motion from the previous window. This yields **smooth, natural transitions** and enables **real-time user control**.
>
> **3. Systematic conditioning redesign and ablation**
> We do not merely add conditions, we decompose and experimentally verify which terms are necessary for the multi-human, long-horizon setting. For example,
>
> - **Local conditions** (target human history, partner history, step index, word-level text) capture fine-grained temporal and social dependencies.
> - **Global conditions** (window index, total sequence length, command-level text) maintain long-term temporal consistency.
>   Ablations show that both are essential, removing either group significantly degrades multi-human long-horizon stability. Such systematic conditioning analysis has not been performed in priorMDM variants.
>
> Additionally, we have compared ComMDM and HINT in our main experiments, as shown in Tab. 1, HINT outperforms ComMDM on all metrics.
>
> **Table.** Comparison with ComMDM on InterDataset and InterX. (already included in Tab1 in the main manuscript)
> |  Dataset   | Method |      R@Top3$\uparrow$       |       FID$\downarrow$       |     MM Dist$\downarrow$     | Diversity$\rightarrow$      |
> | :--------: | :----: | :-------------------------: | :-------------------------: | :-------------------------: | --------------------------- |
> | InterHuman |   GT   |     $0.701^{\pm .008}$      |     $0.273^{\pm .007}$      |     $3.755^{\pm .008}$      | $7.948^{\pm .064}$          |
> |            | ComMDM |     $0.466^{\pm .010}$      |     $7.069^{\pm .054}$      |     $6.212^{\pm .021}$      | $7.244^{\pm .038}$          |
> |            |  HINT  | $\mathbf{0.672}^{\pm .004}$ | $\mathbf{3.100}^{\pm .035}$ | $\mathbf{3.796}^{\pm .001}$ | $\mathbf{7.898}^{\pm .023}$ |
> |   InterX   |   GT   |     $0.736^{\pm .003}$      |     $0.002^{\pm .0002}$     |     $3.536^{\pm .013}$      | $9.734^{\pm .078}$          |
> |            | ComMDM |     $0.236^{\pm .004}$      |     $29.266^{\pm .067}$     |     $6.870^{\pm .017}$      | $4.734^{\pm .067}$          |
> |            |  HINT  | $\mathbf{0.682}^{\pm .003}$ | $\mathbf{0.278}^{\pm .012}$ | $\mathbf{4.007}^{\pm .016}$ | $\mathbf{8.886}^{\pm .066}$ |
>
> ---
>
> **References:**
>
> - [`R1`] Yoni Shafir, et al. Human motion diffusion as a generative prior.  *International Conference on Learning Representations (ICLR)*, 2024.

---

> ### Author Response · Authors · 2025-11-23
> **Reviewer uvFZ (2/2)**
>
> > #### Q2. “How do you control error accumulation across windows?”
>
> Error accumulation across windows can be controlled using the following methods,
> **1. Short-window latent prediction reduces error propagation.**
> Rather than rolling out frame by frame, HINT operates in a canonicalized latent space and predicts future motion in short windows (history length $H$, future length $K$). Each step only propagates errors at the **window level**, and the diffusion process refines a coherent latent trajectory within each window, which empirically stabilizes long-horizon rollouts.
>
> **2. Hierarchical conditions and canonicalized latent space prevent drift.**
> Local conditions enforce **short-term dependencies and fine-grained semantic alignment ** inside each window, while global conditions anchor each window to the overall script, **preventing long-term semantic drift**. In addition, encoding motion in canonicalized per-person coordinates while feeding global geometry as explicit relative transforms decouples global position from motion semantics, reducing the amplification of small pose errors over time.
>
> **3. Autoregressive generation enables interactive correction.**
> Since our model is online, users may issue **light steering commands** (e.g., “move slightly to the right”) to correct deviations, a capability **not available in offline methods**. This is an inherent advantage of an online autoregressive design.
>
> **4. Global goal guided autoregressive generation.**
> If the dataset provides **additional global anchors**, for example, a text prompt such as “go to the bed and sit on the bed’’, then the location of the bed can be supplied in advance as a conditioning term to the diffusion network. This global goal serves as a high-level anchor that guides the local window predictions, ensuring that the generated motion does not drift away from the intended final objective.
>
> Moreover, our long-horizon visualizations further demonstrate the effectiveness of our design(as shown in the supplementary material under `rebuttal_demo/long_seq`).
>
> We have emphasized these points in the revised version.
>
> ---
> We would like to sincerely thank Reviewer `uvFZ` again for the valuable time and constructive feedback provided during this review.

---

> ### Comment · Reviewer_uvFZ · 2025-11-28
>
> Thanks for your rebuttal. However, I can only maintain my score. I believe this article is in a borderline state and requires further decision by the AC.

---

### Author Response · Authors · 2025-11-23
**Rebuttal to All Reviewers**

We sincerely appreciate the valuable work by all ACs and reviewers. We are delighted that HINT is considered to show "**solid quantitative improvements on realism (FID)** [`uvFZ`], **effectiveness for multi-agent motion generation**[`uvFZ`, `ArWa`, `8Sbr`], **clear motivation and practical conditioning design**[`wSXb`], **novelty and effectiveness of hierarchical interaction modeling**[`ArWa`]. We revised the manuscript according to the suggestions, and changes are marked in **blue**.

We summarize the newly added sections as follows:

- **Appendix A:** Details on extending HINT from two-human to multi-human scenarios, with additional experiments.
- **Appendix B.4:** Further details of the experimental setup.
- **Appendix G:** Additional experiments on improving physical plausibility.

Below we try our best to address the common concerns and questions.
### Q1. Scaling from two-human to multi-human motion
Table 1 reports quantitative results on InterHuman and InterX, two standard benchmarks for two-human motion generation. Since there is currently no dataset with variable numbers of agents for training text-guided motion with more than two humans, we do not retrain HINT in multi-person settings. Instead, we directly apply the model trained for two-person generation to scenarios with more agents. Below, we describe this procedure in more detail.

In the two-person case, all conditioning signals are defined with respect to the target agent, and thus remain unchanged when moving to multi-person scenarios, except for the **partner’s history motion**. Concretely, for two agents, we pad the partner’s history motion with zeros to a fixed length and use it as a condition term. For more than two agents, we concatenate all partners’ history motions and then pad the resulting vector. To improve robustness, during training we randomly shift the location of the partner history within the padded vector. Empirically, although HINT is only trained on two-person data, it generalizes reasonably well to multi-person generation (see Fig. 1).

This approach, however, introduces challenges in **role identification** [`wSXb`, `8Sbr`]. Trained on two-person data, the model only needs to distinguish between the **target person** and **the other person**. When the number of agents exceeds two, this assumption no longer holds: the model cannot reliably resolve all pairwise interactions [`8Sbr`], sometimes yielding repetitive behaviors, as observed in Fig. 1 and the supplementary videos.

To address this issue, we extend the HOI-M$^3$ [`R1`] into HOI-M$^{3*}$, enriching it with atomic textual descriptions. HOI-M$^3$ contains multi-human interactions (2–5 agents) in indoor scenes. In total, we annotated 52 motion sequences (approximately six minutes each in duration), comprising 1.1 million frames and 1919 textual descriptions, illustrated in Fig A-2 in Appendix A. HOI-M$^{3*}$ adds sentence-level annotations like “Person0 sits on sofa and talks to Person1,’’ which inherently encodes role identities and can be used to condition the diffusion model.

We finetune HINT on HOI-M$^{3*}$, and the corresponding results are presented in Appendix A. More videos are shown in supplementary materials. The results shows that our framework is structurally capable of handling a variable number of humans.
### Q2. Physical realism, inter-body contact, and penetration
**Physical plausibility** is an important objective in multi-human motion generation. In HINT, we mainly rely on explicitly encoding relative transformations between humans and feeding them as conditions to the diffusion network, enabling the model to learn the physical dynamics of multi-person interactions directly from data. We do **not** impose explicit contact losses or collision penalties, as sufficiently **large and diverse data** can encourage the network to implicitly capture physical regularities, similar to observations in large-scale representation learning (DINOv3 [`R2`]) and video generation systems such as Sora [`R3`].

Nevertheless, our framework is fully **compatible with physics-aware losses**, an additional physics term can be straightforwardly added to the training objective. We further include experiments on InterX (Appendix G), showing that such losses can further reduce slipping and penetration artifacts.

**In summary**, although fine-grained physical interaction modeling is not the primary focus of our work, the framework can **seamlessly incorporate these physical constraints** when needed.

**References:**

- [`R1`] Zhang, Juze, et al. "HOI-M^ 3: Capture Multiple Humans and Objects Interaction within Contextual Environment." *Proceedings of the IEEE/CVF Conference on Computer Vision and Pattern Recognition*. 2024.
- [`R2`] Siméoni, Oriane, et al. "Dinov3." *arXiv preprint arXiv:2508.10104* (2025).
- [`R3`] Liu, Yixin, et al. "Sora: A review on background, technology, limitations, and opportunities of large vision models." *arXiv preprint arXiv:2402.17177* (2024).

---

### Author Response · Authors · 2025-11-29
**Global Response**

We sincerely thank for the valuable time and the constructive feedback by all ACs and reviewers. HINT is, to the best of our knowledge, the first framework that supports text-driven, online multi-human motion generation. We introduce a canonicalized latent space that decouples motion semantics from inter-person geometry, and we further design a hierarchical condition modeling scheme to handle the increased spatiotemporal complexity of multi-human interaction generation. Extensive quantitative and qualitative experiments demonstrate the effectiveness of our approach.

We are delighted that the reviewers consistently recognize that HINT provides **solid quantitative gains in realism (FID)** and **effective multi-agent motion generation** [`uvFZ`, `ArWa`, `8Sbr`], is built on a **clear and practical decomposition of canonicalization and hierarchical conditioning** with **well-motivated design choices** [`uvFZ`, `wSXb`], and offers **novel and effective hierarchical interaction modeling that decouples global positions from individual motion semantics and scales naturally beyond two agents** [`ArWa`, `8Sbr`].

During the rebuttal period, we further validated our framework on the newly reannotated multi-human dataset HOI-M$^{3*}$, providing additional quantitative results and visualizations. In response to reviewers’ concerns regarding physical plausibility, we incorporated penetration-related losses into the original framework and reported quantitative metrics on penetration and foot-sliding. These results demonstrate that our method is naturally compatible with such physical constraints, even though they are not the core focus of our contribution. We also presented longer-horizon generation videos, showing that the combination of our VAE + diffusion design, the canonicalized latent space, and the autoregressive sliding-window mechanism effectively controls error accumulation across windows. We are pleased that these additional analyses were well received by the reviewers and also grateful to reviewer `ArWa` for describing HINT as **a sufficiently elegant and impressive piece of work**.

Regarding the novelty concerns raised by reviewers `uvFZ` and `wSXb`, we would like to clarify that HINT is the **first unified framework** that simultaneously supports text-driven, online, and variable-number human motion generation. Our **ego-centric canonicalized latent space** explicitly decouples individual motion semantics from group geometry, enabling the model to scale naturally to multi-human scenarios without modifying the diffusion backbone. In addition, to address the more complex spatiotemporal dependencies inherent in long-horizon multi-human generation, we designed a **hierarchical condition modeling** scheme and systematically validated the necessity of each condition through **extensive ablations**. Additionally, our method consistently **outperforms** available online baselines (Tab. 1 and Tab. A-1), demonstrating clear advantages. The supplemental experiments further demonstrate that our framework can **seamlessly integrate physical constraints and richer multi-human datasets**, laying the **foundation** for more realistic and complex multi-agent interactions in future work.

We deeply appreciate the reviewers and ACs for their constructive feedback, thoughtful evaluations, and time devoted to our submission, all of which have significantly strengthened the quality of our work. We will also release our code to the community to support further research and ensure reproducibility.

---

### Meta-Review · Area_Chair_EY4u · 2025-12-10

**Summary:**

This paper introduces HINT, an autoregressive diffusion framework that uses canonicalized latent spaces and hierarchical conditions to generate text guided multi human motion with improved realism and long horizon stability. Reviewers agreed that the design is clear and the quantitative gains are solid, though two initially viewed the contribution as incremental and questioned novelty, drift control, and multi person interaction richness. The rebuttal addressed most technical concerns with detailed comparisons, new multi human experiments, and physical plausibility metrics, which led one reviewer to raise their score and another to increase confidence, while two reviewers maintained borderline assessments.

**Reviewer Concerns:**

The rebuttal effectively addressed concerns about drift across windows, the role of word-level guidance, and the feasibility of extending the model to more than two humans. It also responded well to questions about physical plausibility by providing concrete penetration and foot-sliding metrics and clarifying how relative transforms are computed. What remains less resolved are the doubts from two reviewers about the depth of novelty and the still-limited richness of multi human interactions in the qualitative results, which continue to influence their overall assessment.

**Reviewer Scores:**

Reviewer uvFZ began with a score of 4 and stated after the rebuttal that they would keep this score, so no change is expected. Reviewer ArWa began with a 6 and later increased their confidence, indicating they would likely have kept the same score. Reviewer wSXb also started at 4 and explicitly said the rebuttal did not change their overall view, so their score would remain unchanged. Reviewer 8Sbr began at 4 and wrote that they would increase their score after reading the rebuttal, suggesting a move to a positive borderline score.

---

### Decision · Program_Chairs · 2026-01-26

Reject